



# Altered Seasonal Sensitivity of Net Ecosystem Exchange to Controls Driven by Nutrient Balances in a Semi-arid Savanna

Laura Nadolski[1,2], Tarek S. El-Madany[1], Jacob Nelson[1], Arnaud Carrara[3], Gerardo Moreno[4], Richard Nair[5], Yunpeng Luo[6], Anke Hildebrandt[2,7], Victor Rolo[4], Markus Reichstein[1], Sung-Ching Lee[1]

[1]Biogeochemical Integration, Max-Planck Institute for Biogeochemistry, Jena, Germany
[2]Faculty of Chemistry and Earth Science, Friedrich-Schiller University, Jena, Germany
[3]Fundacion Centro de Estudios Ambientales del Mediterráneo (CEAM), Valencia, Spain
[4] Faculty of Forestry, Institute of Dehesa Research (INDEHESA), Universidad de Extremadura, Plasencia 10600, Spain
[5]Discipline of Botany, School of Natural Sciences, Trinity College Dublin, Dublin Ireland
[6]Swiss Federal Institute for Forest, Snow and Landscape Research WSL, 8903 Birmensdorf, Switzerland
[7]Department Computational Hydrosystems, Helmholtz Centre for Environmental Research (UFZ), Leipzig, Germany

*Correspondence to*: Laura Nadolski (lnadolski@bgc-jena.mpg.de)

**Abstract**

Semi-arid ecosystems dominate variability and trend of the terrestrial carbon sink. They are sensitive to environmental changes following anthropogenic influence, such as an altered ratio of nitrogen (N) to phosphorus (P) due to increasing N deposition. Semi-arid savannas with different vegetation compositions have complex carbon dynamics, and their responses to environmental change are not yet well understood. We analysed a long-term (2016-2022/2023) dataset of flux, biometeorological and vegetation data (satellite and ground measurements) of a manipulated semi-arid savanna to reveal how altered nutrient levels and stoichiometric balance affect the seasonal sensitivity of net ecosystem exchange (NEE) to its drivers. We used the Singular Spectrum Analysis to extract the seasonal signal of all variables and assessed the key drivers of NEE over the study period as a whole and in different seasons, using Pearson correlation and Information Theory. We found that both N and N+P addition to the ecosystem increased seasonal NEE variability, driven by greenness of the herbaceous layer. Analysing 7 years of data together, the water limitation in summer and energy limitation in winter outcompeted the fertilization effect. By investigating different phenological seasons, effects of nutrient addition on NEE-control relationships became clearer. In the summer, N+P addition led to a potential change in species composition and productivity resulting in a stronger interaction between herbaceous layer and NEE. During the transitional seasons (i.e., drydown and regreening), which determine the senescence and regreening of the herbaceous layer, we found NEE to be less sensitive towards meteorological drivers like relative humidity, radiation and air temperatures with N addition. The increasing NEE variability might become even more pronounced with N deposition and a changing climate in the future.





## 1 Introduction

Terrestrial ecosystems are a major component of the global carbon cycle, with the ability to store significant amounts of carbon (Friedlingstein et al., 2022). While forests and wetlands contribute most to the terrestrial carbon sink, semi-arid ecosystems dominate its trend and interannual variability (Ahlström et al., 2015; Poulter et al., 2014; Zhang et al., 2016). Semi-arid ecosystems typically take up carbon from the atmosphere during the wet season and are dormant or emit carbon during the dry season (Metz et al., 2023). Net ecosystem exchange (NEE) describes this balance between carbon uptake through photosynthesis, typically expressed as gross primary productivity (GPP), and carbon release through ecosystem respiration ($R_{eco}$). NEE in semi-arid regions varies strongly from year to year, depending on the climatic conditions and water availability (Haverd et al., 2017; Piao et al., 2020).

Despite their important role in the global carbon cycle, semi-arid ecosystems and their dynamics are still not well understood. Long-term in-situ measurements from these regions are scarce. Particularly, eddy covariance (EC) measurements, which provide high-frequency and continuous ecosystem trace gas and water flux data (Baldocchi, 2020), are underrepresented in these regions (Jung et al., 2020). Consequently, semi-arid ecosystems remain poorly represented in terrestrial biosphere models (Fawcett et al., 2022; MacBean et al., 2021) due to their complex structure and high spatio-temporal variability, which are difficult to generalize.

Recently efforts have been made to reveal drivers of NEE in semi-arid savannas to understand better their role in the global carbon cycle (Baldocchi and Arias Ortiz, 2024; Kannenberg et al., 2024; Ma et al., 2007, 2016; Zhang et al., 2010). Water related variables like precipitation and soil moisture availability are amongst the main NEE drivers (Archibald et al., 2009; Baldocchi and Arias Ortiz, 2024; Del Grosso et al., 2018; Huang et al., 2016b; Morgan et al., 2016), as they promote plant photosynthesis (Parton et al., 2012) and enhance heterotrophic respiration rates (Ma et al., 2016). Furthermore, photosynthetically active radiation (PAR), vapor pressure deficit (VPD) and air temperatures can strongly impact NEE (Archibald et al., 2009; Baldocchi and Arias Ortiz, 2024; Del Grosso et al., 2018). Light absorption propels the electron transport mechanism integral to photosynthesis. VPD influences the modulation of stomata opening and temperature impacts the kinetics of enzymes involved in carboxylation processes, the process of carbon uptake into the plant (Baldocchi and Arias Ortiz, 2024). Also, other biotic factors, like soil microbial communities and organic matter play an important role in the ecosystem carbon cycle and contribute to $R_{eco}$ (Austin and Vivanco, 2006; Bastida et al., 2016; Hu et al., 2014). These drivers can differ for different vegetation types. Typical ecosystems in semi-arid regions are savannas where coexisting vegetation layers (e.g., tree and grass) interact in complex ways. The layers differ in their rooting depths (Moreno et al., 2005; Rolo and Moreno, 2012), water use strategies (Cubera and Moreno, 2007; Miller et al., 2010; Steiner et al., 2024) and phenological and life cycle strategies (Whitecross et al., 2017). Especially the herbaceous vegetation in two-layer ecosystems is often underestimated in its importance for the ecosystem water and carbon fluxes (Dubbert et al., 2014). In the Iberian Peninsula *dehesas* (or *montados* in Portugal), human shaped savanna-like agroecosystem, are wide-spread (Den Herder et al., 2017). *Dehesas* are open oak woodlands with an herbaceous layer that consists mainly of annual grasses and sometimes crops. The





tree layer is evergreen (Moreno, 2008), whereas the herbaceous layer typically follows an annual cycle of growth, senescence
and regreening (Ma et al., 2007; Perez-Priego et al., 2015). As savannas are typically characterized by changing resource
limitations throughout the year (Luo et al., 2020; Ries and Shugart, 2008), limited by water in the dry season and by nutrients
and energy in the wet season (Moreno, 2008; Morris et al., 2019; Nair et al., 2019), these drivers change with the seasons
throughout the year. The complex interactions between tree and grass layers, along with changing limitations result in a high
complexity of the ecosystem's carbon dynamics, which remain to be fully understood.
Semi-arid ecosystems face numerous human-induced environmental changes, including stoichiometric imbalances between
nitrogen (N) and phosphorus (P). These imbalances arise from increasing N inputs into ecosystems due to fertilizers and
combustion of fossil fuels (Steffen et al., 2015) without corresponding increase in P inputs (Penuelas et al., 2013). Few studies
have dealt so far with the impact of altered nutrient levels on NEE and its drivers in semi-arid regions. The availability and
stoichiometric balance of N and P influences ecosystem functioning and plant traits (Reichstein et al., 2014), water use
efficiency (El-Madany et al., 2021; Huang et al., 2016a), canopy structure (Migliavacca et al., 2017), composition of species
(Sardans et al., 2012), and the seasonality of vegetation activity (Luo et al., 2020). However, different plant types react
differently to changes in nutrient availability, due to variations in generation times and buffering capacities (Pardo et al., 2011).
Therefore, the understanding of the response of complex tree-grass ecosystems to changes in N and P availability and their
stoichiometric balance is still poor.

In this study we took advantage of the unique long-term dataset collected in a semi-arid *dehesa,* Majadas de Tiétar, in South-
Western Spain. A large-scale nutrient addition experiment has been running here since 2015, providing an exceptional
opportunity to study the long-term influence of altered N:P ratios on ecosystem functioning (El-Madany et al., 2021). Three
EC flux towers have been set up, with the footprint of one tower receiving N fertilization, another one receiving N+P
fertilization and the third serving as control. Previous studies found that both treatments increased the annual carbon uptake of
the ecosystem and that N+P addition increased the water use efficiency of the ecosystem more than N-only addition, which
could be attributed to higher transpiration rates and a changed root strategy in the N-only fertilized plot (El-Madany et al.,
2021; Nair et al., 2019). Nutrient addition further led to a higher seasonal amplitude of maximum GPP and a faster increase
during the regreening period, but also a faster senescence during the drydown period, which indicates changes in plant structure
and physiology (Luo et al., 2020).
Here we analysed a 7-year (2016-2022) timeseries of daily values of environmental and biogenic variables from Majadas de
Tiétar, combining flux data, meteorological measurements, digital repeat photography and satellite data to address the
following questions: How do altered nutrient levels and stoichiometric balance affect

94        –    annual NEE and its variability in a semi-arid savanna?
95        –    the relationship between NEE and its key controls?
96        –    the relationship between NEE and its key controls in different seasons?
97        –    the sensitivity of NEE to its controls over time?



The relationships between NEE and its controls vary across different time scales (Mahecha et al., 2007). To disentangle these
timescales from a time series and eliminate noise from the high-frequency measurements, we can use decomposition methods
(Linscheid et al., 2020). On short timescales the NEE sensitivity follows the diurnal cycle of the sun, showing a great
dependency on radiation. Ecosystem-level responses, in contrast, often develop on scales of months, seasons or years (Ma et
al., 2016). Therefore, we extracted the seasonal signals of all variables from the timeseries with the Singular Spectrum Analysis,
a data-driven timeseries decomposition method. On the seasonal scale we assessed the key drivers of NEE with Pearson
correlation coefficient and information theory-based methods, accounting for collinear relationships as well as leading and
lagging effects. As NEE controls vary in their importance throughout the year due to a high seasonality of the ecosystem, we
split the dataset into phenological seasons defined by vegetation responses to different limitations.

## 2 Material and Methods

### 2.1 Site Description

The Majadas de Tiétar research site is located in Western Spain (39°56′25″N 5°46′29″W). The local ecosystem consists of an
herbaceous stratum and scattered evergreen oak trees (98% *Quercus ilex*). The tree density is around 20-25 trees per hectare
(El-Madany et al., 2018), the fractional canopy cover of trees is 23 % and the canopy height is on average 8.7 m (Bogdanovich
et al., 2021). The tree leaf area index (LAI) is around 0.35 $m^2\,m^{-2}$, the grass layer has a peak LAI in spring but is quite spatially
heterogeneous at between 0.5 and 2.5 $m^2\,m^{-2}$ (Migliavacca et al., 2017). The site is managed and continuously used for grazing
livestock at a low density of 0.3 cows per hectare (El-Madany et al., 2018). In the driest months (July - September) the farmers
move the cattle to nearby mountain grasslands (personal communication).
The climate at the site is semi-arid with an annual precipitation of around 650 mm with strong interannual variability. Almost
85 % of the annual precipitation falls in the wet season between October and April, whereas the rest of the year is dry with
occasional rains (El-Madany et al., 2021). According to Nair et al. (2024) we defined five different seasons. Spring is the main
growing season and usually starts around March and ends in late May. Then the drydown period starts and the grasses start to
become senescent due to depletion of soil moisture, increasing temperatures, radiation and vapor pressure deficit. The summer
(typically between end of June until end of September) is characterized by long-lasting dryness and a dormant/dead herbaceous
layer. With the onset of precipitation (usually in October), the autumn starts and the herbaceous layer regreens (Nair et al.,
2024). The winter months (December-February) are energy limited. The onset and offset of the different seasons vary from
year to year, depending on water availability and winter temperature (Luo et al., 2020). The mean annual temperature is 16.7°C
with an average minimum temperature of -4.7°C and maximum temperature of 41.1°C (between 2004-2019) (El-Madany et
al., 2021). Dominant wind directions are West-Southwest and East-Northeast (El-Madany et al., 2018).
Three EC towers at ecosystem level were operated simultaneously at the site. The ecosystem is heterogeneous with a high
variability in plant species in the herbaceous layer (at scale of centimetres) and tree cover (at scale of meters). It becomes
homogeneous on the scales of a few hundreds of meters. The daytime flux footprints of the three towers correspond to the





scale being homogeneous and they do not overlap with each other under prevailing meteorological conditions (El-Madany et
al., 2018). The control tower (Fluxnet ID: ES-LMa) has been operated since 2003, it is hereafter referred to as CT. The North
tower (Fluxnet ID: ES-LM1) was set up at a distance of 450 m from CT in north-western direction and the South tower (Fluxnet
ID: ES-LM2) was located 630 m in southern direction from CT (El-Madany et al., 2018). Since 2015 a large-scale fertilization
experiment has been conducted at the site, where N fertilizer is added in the footprint of the North tower (hereafter referred to
as NT) and N and P fertilizer are applied in the footprint of the South tower (referred to as NPT) (El-Madany et al., 2021).
After the initial application of 100 kg N ha$^{-1}$ and 50 kg P ha$^{-1}$ in March 2015 and November 2014, respectively, additional N
and P addition were applied every 1-2 years with lower doses. Next to each flux tower there is a radiometric tower setup,
measuring radiation components above tree and grass layer, alternating every 15 minutes providing half-hourly measurements
for each layer.

**2.2 Eddy Covariance and Biometeorological Data**

Each of the three EC towers continuously measures sensible heat (H), latent heat (LE) and $CO_2$ flux. Each system is equipped
with a R3-50 sonic anemometer (Gill Instruments Limited, Lymington, UK) to measure three-dimensional wind components
and sonic temperature, and a LI7200 infra-red gas analyzer (Licor Bioscience, Lincoln, Nebraska, USA) to measure $CO_2$ and
$H_2O$ mixing ratios. The measurement heights are the same at 15 m above ground (El-Madany et al., 2021). The flux and
meteorological data were collected as described by El-Madany et al. (2018). The raw high-frequency data was processed with
EddyPro v.7.0.9 (Fratini and Mauder, 2014). The post-processing was done in R using the REddyProc package (Wutzler et al.,
2018). The storage corrections of the $CO_2$-flux were made with profile measurements from seven points on the flux towers. A
friction velocity (u*) threshold was applied following Papale et al. (2006) and data with u* values below the defined threshold
were removed. Missing and bad quality data were gap-filled (Mauder and Foken, 2011; Reichstein et al., 2005) for calculating
the annual budgets. Additional atmospheric variables that we used are air temperature (Ta) and relative humidity (Rh)
measured at two heights (2 m and 15 m), vapor pressure deficit (VPD), $CO_2$-flux (NEE), air pressure (air_press) and friction
velocity (ustar). Furthermore, we incorporated radiometric components such as longwave downward radiation (LWDR),
shortwave downward radiation (SWDR) and photosynthetically active radiation (PAR). Soil measurements comprised soil
water content integrated over the top 20 cm of the soil (SWCn), soil temperature in open pasture ($T_{soil}Sun$) and below oak tree
canopy ($T_{soil}Shd$), and soil heat flux in open pasture (SHF_Sun) and below oak tree canopy (SHF_Shd).
Some small data gaps existed in meteorological variables and were filled with the average of the remaining two towers and
interpolation. However, the PAR sensor at CT had a malfunction over a long period and therefore we used the PAR timeseries
from NT to substitute, as the incoming radiation should not differ substantially between the towers (El-Madany et al., 2018).
In addition, we calculated Evaporative Fraction (EF) as the ratio between LE and available energy (EF = LE/(LE+H)) (Gentine
et al., 2007; Tong et al., 2022). EF is a normalized measure of the surface energy partitioning and can serve as diagnostic of
vegetation water status (Nutini et al., 2014). We calculated it at half-hourly timesteps from only positive LE and H values to
not introduce extreme outliers into the analysis. EF is strongly linked to meteorological variables like soil moisture, VPD and



net radiation (Gentine et al., 2007; Tong et al., 2022), but also to vegetation cover and LAI (Gentine et al., 2007). A full
overview of analysed variables is shown in Table 1.

**2.3 Vegetation indices**

We used three different vegetation indices to represent vegetation greenness, derived from in-situ data (Green chromatic
coordinates and albedo) and satellite data (Normalized difference vegetation index).

Green chromatic coordinate (GCC) is an effective measure for describing greenness variation in semi-arid ecosystems (Luo et
al., 2018, 2020). We used daily mean GCC values extracted from the RGB images collected by digital cameras (Stardot
NetCam 5MP) which were installed at the top of each ecosystem EC tower facing north, collecting images every 30 minutes.
The cameras were set up according to the protocol of the PhenoCam network (https://phenocam.sr.unh.edu/webcam/tools/)
and collect red, blue, green (RGB) images (Luo et al., 2018). GCC was computed as the fraction of green digital numbers
$(G_{DN})$ in relation to the sum of red $(R_{DN})$, blue $(B_{DN})$ and green digital numbers (Richardson et al., 2009):
$$GCC = \frac{G_{DN}}{R_{DN} + B_{DN} + G_{DN}} \tag{1}$$


At each site we selected two regions of interest in which we calculate GCC, one capturing the grass layer (gcc_gr) and one
capturing the trees (gcc_tr). The data derived from RGB images can be found on the website of the PhenoCam network (IDs:
ES-LM1, ES_LMa and ES_LM2 for the NT, CT and NPT, respectively). At each site we chose the masks GR_1000 for the
grass layer and EB_1000 for the trees.

We further calculated albedo as the ratio of outgoing shortwave radiation to incoming shortwave radiation, measured at the
radiometric tower setup at each site. We distinguished ecosystem albedo (Alb_eco), tree (Alb_tr) and grass albedo (Alb_gr) to
account for reflectance and as another proxy for vegetation greenness and water status of the plants. We used daily averages
from only daytime hours (11:00-15:00) to guarantee a high solar zenith angle for reliable measurements. Furthermore, cloudy
days were filtered out and only timesteps where the ratio of downward radiation to extra-terrestrial radiation at the top of the
atmosphere was 0.7 or more were kept (Wood et al., 2015).

Finally, we use Normalized Difference Vegetation Index (NDVI) from the FluxnetEO dataset, as a proxy describing the amount
and health of vegetation cover (Tucker, 1979). The dataset complements ground measurements by providing satellite-based
vegetation indices, surface reflectance and land surface temperatures for a 2 km radius around a flux site (Walther et al., 2022).
We use NDVI from MODIS (Moderate Resolution Imaging Spectroradiometer) with a daily temporal resolution (Walther et
al., 2022). NDVI is calculated from the normalized difference between the reflectance of near-infrared (NIR) and red-light
bands (Tucker, 1979).



**Table 1: Flux, meteorological, soil variables and vegetation indices used in this study. Soil heat flux and soil temperatures were calculated based on the shadow fraction estimated from the solar zenith angle (variable SZA) and a canopy cover of 20%.**

| variable name | variable description | unit | measurement device | measurement height/ depth |
|---|---|---|---|---|
| **NEE** | net ecosystem exchange on ecosystem level | μmol m-2 s-1 | R3-50, Gill LTD UK, LI-7200 | 15m, 15.5m (CT) |
| **EF** | evaporative fraction | | R3-50, Gill LTD UK, LI-7200, calculated | 15m, 15.5m (CT) |
| **air_press** | air pressure | Pa | Young 61302V | |
| **Rh02** | relative humidity at 2m | % | CPK1-5 | 2m |
| **Rh15** | relative humidity at 15m | % | CPK1-5 | 15m |
| **Ta02** | temperature at 2m | degreeC | CPK1-5 | 2m |
| **Ta15** | temperature at 15m | degreeC | CPK1-5 | 15m |
| **VPD** | water vapor pressure deficit | Pa | calculated | 15m |
| **ustar** | friction velocity | m s-1 | R3-50, Gill LTD UK | 15m |
| **SWDR** | short wave downward radiation | W m-2 | CMP22/CNR4 | 9m |
| **LWDR** | long wave downward radiation | W m-2 | CNR4 | 9m |
| **PAR** | incoming photosynthetically active radiation | umol m-2 | Kipp& Zonen PQS1 | 9m |
| **SWCn** | normalized soil moisture content for top 20cm | | ML2x, Delat-T Devices Ltd | 20cm |
| **SHF_Sun** | soil heat flux sun | W m-2 | HP3/CN3 Rimco | 5cm |
| **SHF_Shd** | soil heat flux shadow | W m-2 | HP3/CN3 Rimco | 5cm |
| **$T_{soil}$Sun** | soil temperature sun | degree C | UMS Th3-s | 10 cm |
| **$T_{soil}$Shd** | soil temperature shadow | degree C | UMS Th3-s | 10 cm |
| **Alb_eco** | ecosystem Albedo | | CNR4 | 15 m |
| **Alb_gr** | grass level Albedo | | CNR4 | 9m (CT, NT), 12m (NPT) |
| **Alb_tr** | tree level Albedo | | CNR4 | 9m (CT, NT), 12m (NPT) |
| **gcc_gr** | grass level green chromatic coordinates | | stardot netcam SC5 | 15m |
| **gcc_tr** | tree level green chromatic coordinates | | stardot netcam SC5 | 15m |
| **NDVI** | normalized difference vegetation index | | MODIS satellite | |





200

## 2.4 Data Analysis

### 2.4.1. Aggregation to daily data

For our analysis we calculated from the biometeorological and flux data daily mean values aggregated from the half-hourly measured values during daytime. Daytime includes only values measured after sunrise and before sunset, identified using the *suncalc* package in R (Thieurmel, 2017). We discarded flux measurements with quality flag 1-3 and kept only measured values to not confound following analyses of NEE controls with gap-filled values which are based on other meteorological variables. This does not apply to vegetation indices as they were calculated as described above. GPP and $R_{eco}$ were not assessed in this study as partitioning methods depend on other environmental factors that would also confound the analysis of NEE controls. If not stated differently, the following analyses cover the 7-year period from 2016-2022 as in this time all variables are available. For the assessment of NEE variability and budgets, we utilized data spanning 8 years (2016-2023) because this extended dataset was available and incorporating additional years enhances the robustness of observed trends.

### 2.4.2 Time Series Decomposition with Singular Spectrum Analysis

Decomposition methods assume that observed time series are composed of additively superimposed sub-signals, each shaped by different scales of variability (Mahecha et al., 2010). Consequently, the time series represents the sum of a trend, oscillatory components at various scales, and noise (Liu et al., 2022).

Here we used Singular Spectrum Analysis (SSA) for the decomposition. SSA is entirely data-driven and non-parametric and is therefore free of the bias of function-selection (Golyandina et al., 2001; Liu et al., 2022; Mahecha et al., 2007). This makes it advantageous compared to other decomposition methods like Fourier and wavelet analysis (Baldocchi et al., 2021). It is more flexible in grouping components of similar frequencies than wavelet decomposition (Liu et al., 2022) and able to detect aperiodic or non-harmonic sub-signals from short and noisy signals (Golyandina and Zhigljavsky, 2013; Mahecha et al., 2007). Since it is fully phase-amplitude modulated, and relatively robust against instationarities of the signal mean and variance, it is suitable for nonstationary signals (Allen and Smith, 1996; Golyandina and Zhigljavsky, 2013; Yiou et al., 2000). Even fragmented timeseries can be handled with it, as SSA can be used for filling gaps according to the first reconstructed component which is a low-frequency signal (Kondrashov and Ghil, 2006). This makes it particularly useful for flux data (Mahecha et al., 2007).

SSA consists of four steps: embedding, decomposition, grouping and reconstruction. In the first step a one-dimensional timeseries y(t) is embedded into a two-dimensional lagged matrix X, by shifting a moving window of a certain window length (L) along the timeseries. In the second step X is decomposed into its orthogonal components by determining eigenvalues and eigenvectors corresponding to principal components (singular value decomposition). Then the eigenvalues of the covariance matrix X · X are ranked. In the next step, the components are grouped, as some sub-signals consist of a set of components with





complementary oscillatory frequency. In the last step, by inverting the ranked principal components, the reconstructed
components of the original time series are computed. These reconstructed components show how much of the variability of
the original timeseries is associated with the different timescales. A more detailed description of the method can be found in
Golyandina et al. (2018).
Here we used the rssa – package in R (Golyandina and Korobeynikov, 2014) for our analysis. To support our hypothesis that
daily-scale NEE variations are predominantly influenced by radiation, with a neglectable effect of nutrient addition, we
conducted a preliminary analysis extracting the daily signal of NEE and all potential driving variables from half-hourly
measurements. Detailed procedures and results of this analysis are provided in the Supplementary Material (S1).
For our analysis we extracted the seasonal signal of the daily timeseries of all variables shown in Table 1. First, we gap-filled
the timeseries with the igapfill – function. For gap-filling, as a window length (L) of n/2.5 is recommended (Mahecha et al.,
2007), we selected a gap-filling window length of L = 1000 for 2557 datapoints from 7 years of daily data. By conducting a
sensitivity analysis, we found that adding a three-month margin at the beginning and the end of a timeseries can help to reduce
edge effects during the gap-filling (details see S.2).
To extract the seasonal signal, we reconstructed the components of the frequency bin of 15 to 366 (days). We selected L = 732
(2 years) based on the criteria that L should be less than n/2 and ideally an integer multiple of the period length to be extracted
to ensure a clear signal (Biriukova et al., 2021; Golyandina and Zhigljavsky, 2013). Frequency contributions of less than 0.2
were defined as noise (Liu et al., 2022). For the grouping we used the automated method provided by the rssa-package, which
identifies groups using a hierarchical clustering algorithm based on the w-correlation matrix. The w-correlation matrix shows
the weighted correlations between reconstructed components (Buttlar, 2014; Golyandina and Korobeynikov, 2014).
For analysing the changes in seasonal NEE variability and budgets, we used data from 2016-2023. Accordingly, L was set at
1169 (L = n/2.5, with n = 3105). To account for seasonal variability, we calculated for each year the standard deviation of the
reconstructed NEE signal to capture the variation amplitude.

### 2.4.3 Pearson Correlation Coefficient

To find out the key drivers of NEE we first computed in R the Pearson correlation coefficients (*r*) between NEE and all the
investigated variables (Table 1) from the reconstructed seasonal signal using the daily datasets. It is calculated as follows:
$$r = \frac{\sum_{i=1}^{n}(x_i-\underline{x})(y_i-\underline{y})}{\sqrt{\sum_{i=1}^{n}(x_i-\underline{x})^2 \sum_{i=1}^{n}(y_i-\underline{y})^2}} \qquad (2)$$
With n as the timeseries length, $x_i$ and $y_i$ as the single timestep values within the timeseries and $\underline{x}$ and $\underline{y}$ as the sample means.
We calculated values for each tower and then ranked r according to their absolute value to identify the main drivers of NEE.





**2.4.4 Information Theory**

To consider collinear relationships and potential lagging effects between NEE and its controls, we extended our analysis using information theory. Metrics of Mutual Information (MI) are a powerful tool to understand non-linear and feedback-driven relationships in complex ecosystems (Chamberlain et al., 2020; Knox et al., 2018). MI is a non-parametric method and can disentangle interactions on different scales (Chamberlain et al., 2018; Knox et al., 2018; Sturtevant et al., 2016), by describing the average tendency for joint states of two variables X and Y to co-occur (Fraser and Swinney, 1986). This means it quantifies the amount of information that two variables X and Y hold in common, or the reduction of uncertainty of one variable, given the knowledge of the other (Chamberlain et al., 2020; Knox et al., 2021). It is a normalized measure of the statistical dependence of Y on X and no prior knowledge about their relationship is needed (Liu et al., 2022). Larger values indicate higher dependence, or a stronger interaction between the variables. With Shannon entropy ($H_x$) we can quantify the uncertainty in a system:

$$H_x = -\sum_{x_t} p(x_t) log_2\, p(x_t) \tag{3}$$

with p(x) as the marginal probability distribution of X, and $X_t$ as the different states of X in the timeseries t. Here we discretized the states of continuous variables into ten fixed-interval histogram bins, as Sturtevant et al. (2016) and Ruddell and Kumar (2009) showed that ten histogram bins ensure sufficient resolution for a robust estimate. MI then was calculated with both the marginal and joint probability distributions of X and Y, p(x,y):

$$MI = \sum_{x_t} p(x_t, y_t) log_2 \frac{p(x_t, y_t)}{p(x_t) p(y_t)} \tag{4}$$

To make the MI between NEE and different potential drivers comparable, we used here a normalized form of MI:

$$MI_{sync} = \frac{MI}{H_y} \tag{5}$$

We refer to this relative MI as synchronous MI ($MI_{sync}$), as it depicts the interaction between X and Y at the concurrent time step. A further power of MI lies in its capability to account for the temporal direction ($\tau$) of the interaction between X and Y (Liu et al., 2022):

$$MI_{max} = MI_{sync_{(\tau)}} = \frac{\sum_{x_{t-\tau}} \sum_{y_t} p(x_{t-\tau}, y_t) \frac{p(x_{t-\tau}, y_t)}{p(x_{t-\tau}), p(y_t)}}{-\sum_{y_t} p(y_t) p(y_i)} \tag{6}$$

//doi.org/10.5194/egusphere-2024-3190





Positive and negative values of τ show an asynchronous interaction between X and Y, with a lag or lead in Y relative to X. We
chose 60 days as maximum value for τ to check if the potential driving variable (Y) is leading NEE (X) or vice versa (Liu et
al., 2022). We then picked the highest MI value ($MI_{max}$) in this window and the respective day of its occurrence. If $MI_{sync} >$
$MI_{max}$, the interaction is synchronous, if $MI_{sync} < MI_{max}$, the interaction is asynchronous. If τ < 0, Y lags X, if τ > 0, Y leads X
and can therefore be characterized as a driver or control of X. Significance thresholds were calculated from the 95[th] percentile
(p < 0.05) of 1000 Monte Carlo random walks of the independent variable (Chamberlain et al., 2020; Ruddell and Kumar,
2009). We calculated MI measures and the confidence thresholds in R, based on functions by Chamberlain et al. (2020).
We determined $MI_{sync}$ and $MI_{max}$ for the 7-year time series (2016-2022) from the reconstructed seasonal signal. Gap-filled
timesteps by SSA were removed before both the calculation of *r* and MI measures. Only for NDVI we kept them, as the gap-
filling is based on the original timeseries and does not depend on other variables (Walther et al., 2022). It therefore does not
confound the analysis of potential drivers.

**2.4.5 Phenological Seasons**
As the NEE controls vary in their importance in different seasons (Baldocchi and Arias Ortiz, 2024), we calculated $MI_{sync}$ for
each season to better capture how the nutrient addition and stoichiometric balance change the importance of different drivers
over the study period. As this ecosystem's strong seasonality is reflected in vegetation activity, we assigned seasons using
PhenoCam imagery. We defined phenological seasons following Nair et al. (2024). Phenological transition dates were
extracted using GCC at all three sites according to changes between stationary and rising or declining greenness (Luo et al.,
2018). Then, phenological transition dates averaged across the three sites for each year were calculated. According to these
dates, each day of the 7-year timeseries was assigned to one season, describing different phases of net vegetation activity (i.e.,
spring, drydown, summer, autumn and winter, as described above in Chapter 2.1.). Figure 1 illustrates a typical annual cycle
of the seasons at Majadas de Tiétar.
We calculated $MI_{sync}$ values for each pair of interest (NEE and potential driving variable) in each season across all 7 years
together. In addition, we estimated yearly $MI_{sync}$ for each single season (35 datapoints) to evaluate how sensitivity of NEE to
drivers developed over time. To isolate the fertilization effect on the importance of different drivers for NEE, we calculated
the differences in the $MI_{sync}$ values of each season in each year between the fertilized plots and the control plot, i.e., NT – CT
and NPT – CT, referred to as $MI_{diff}$. We plotted the $MI_{diff}$ values for each season along the 7-year period and calculated linear
regressions to confirm whether there are significant trends in the importance of drivers. The significance level was set at p <
0.05. Variables with $MI_{max} < 0.2$ were discarded.





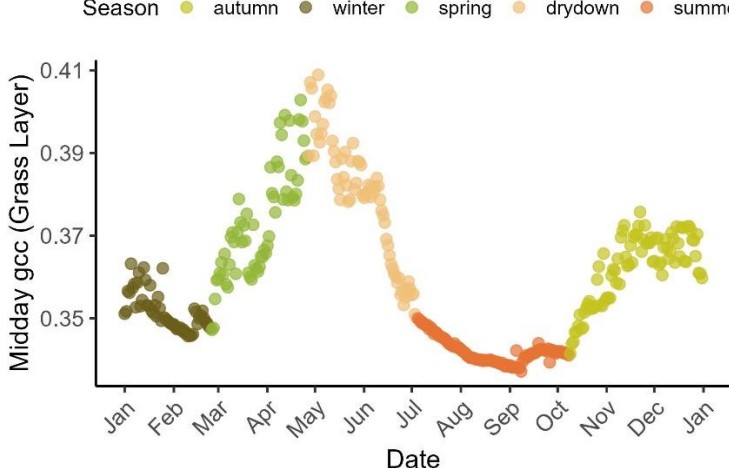


**Figure 1: A typical annual cycle of midday green chromatic coordinates (gcc) derived from the grass layer at the control plot in 2018, showing the five phenological seasons – winter, spring, drydown, summer and autumn. Spring is the main growing season (first peak in May), the grasses become senescent during drydown and dormant in summer, regreening starts (second peak around November) in autumn with the onset of rains, and winter is radiation and temperature limited.**

## 3 Results

### 3.1 Seasonal NEE Variability

At CT, not experiencing any manipulation, the annual ecosystem NEE derived from EC measurements was positive for the 2016-2023 period, with an average annual NEE budget of $90.8\pm 48.0$ gC m$^{-2}$ y$^{-1}$. This indicates that the ecosystem acted as a $CO_2$ source. With fertilization treatment, the measured ecosystem NEE shifted towards $CO_2$ neutrality, with annual averages of $34.1\pm 66.7$ gC m$^{-2}$ y$^{-1}$ and $23.1\pm 69.5$ gC m$^{-2}$ y$^{-1}$ at NPT and NT, respectively. Annual NEE budgets fluctuated between positive and negative values at the fertilized plots, while CT consistently showed positive NEE throughout the years. In the years of 2017, 2022 and 2023, we observed high positive NEE values (i.e., stronger $CO_2$ source) at all three plots. Conversely, in years such as 2016, 2018, and 2021, fertilized areas exhibited higher $CO_2$ uptake, acting as stronger $CO_2$ sinks (Fig.2). This illustrates the high interannual variability of the $CO_2$ fluxes in this ecosystem and the substantial impact of fertilization. Additionally, the nutrient addition led to higher seasonal variability of NEE, as shown by the greater yearly standard deviation of the seasonal reconstructed signal. The variability at NT and NPT further exhibited an increasing pattern over time (Fig.2). In 2017, NEE had comparatively low seasonal variability at all three sites, which might be attributed to the extraordinary dryness in that year.



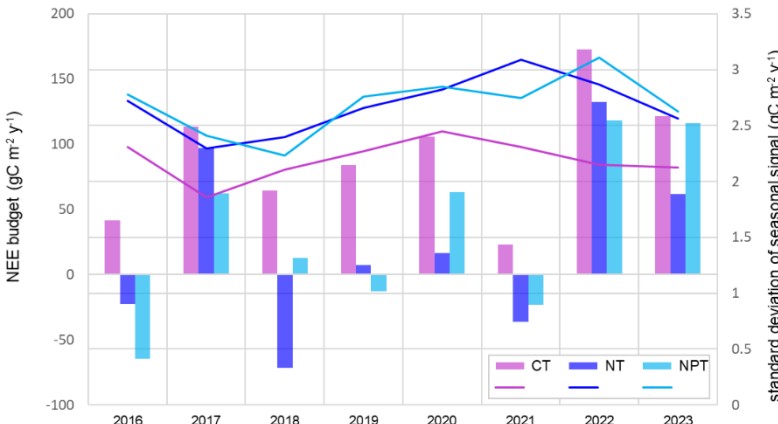

**Figure 2: Annual net ecosystem exchange (NEE) budgets (gC m$^{-2}$ y$^{-1}$) and yearly standard deviations calculated using the seasonal reconstructed signals at the three sites for the 2016-2023 period. CT- control site, NT – nitrogen fertilized site, NPT – nitrogen and phosphorus fertilized site.**

## 3.2 Key NEE controls

We identified key controls of NEE at the three plots, comparing the results of Pearson correlation coefficient *r*, which considers only linear relationships between variables, and with Mutual Information (MI), which accounts for collinear relationships. and MI$_{sync}$ and *r* values show synchronous relationships, MI$_{max}$ values can account for leading and lagging interactions by identifying the day of the highest interaction between the potential driver and NEE within a 60-day window.

At all plots, gcc_gr (i.e., grass layer GCC) and NDVI (at ecosystem level) were the most important predictors of NEE (Fig. 3). Both *r* and MI identified these proxies representing vegetation greenness as the most significant drivers. They were followed by EF (i.e., the fraction of heat transport that is done by LE), which is influenced by meteorological variables (such as soil moisture, net radiation and VPD) as well as vegetation properties like LAI.

Regarding micrometeorological variables, at CT, T$_{soil}$Shd and Ta15 exhibited strong interactions with NEE using both *r* and MI$_{sync}$. Using *r*, LWDR, and SWCn were also important in explaining NEE variations, while MI$_{sync}$ indicated that Ta02 and T$_{soil}$Sun were more significant (Fig.3 (a), (b)). Variables describing water availability, such as VPD, SWCn and Rh were ranked in the middle ranges by MI$_{sync}$ (e.g., VPD at rank 9 and SWCn at rank 14). The MI analysis provided deeper insights into the interactions between the environment and NEE by considering leading and lagging effects, as shown by MI$_{max}$ (Fig. 3(c)). NDVI showed the highest interaction with NEE at a time lag of 16 days, and the second most important driver, gcc_gr, had a lag of 7 days. When considering leading and lagging effects, EF became relatively less important with a value of 0.29 at 15 days. Soil temperatures were identified amongst the five most important controls by MI$_{max}$, with lags of 37 and 43 days for T$_{soil}$Shd and T$_{soil}$Sun, respectively. SWCn was also important (ranked 6$^{th}$) with a 20-day lag. Other variables such as air temperature and VPD showed the highest interaction with a lag of around a month. Radiation-related variables like PAR and





SWDR exhibited long lag times in their highest interaction with NEE (60 days and 53 days, respectively). All MI values can
be found in the Supplementary Material (S3).
At NT, the two synchronous methods agreed on the twelve most important predictors, with the exception that $MI_{sync}$ identified
SWCn and LWDR as less important and $T_{soil}Sun$ as more important compared to $r$ (Fig.3 (d), (e)). Soil temperatures, VPD,
SWCn and air temperatures were among the most significant controls, following the most important drivers, vegetation
greenness and EF. NDVI showed the highest interaction with NEE with a lag of 12 days, followed by gcc_gr with a lag of 6
days. Soil temperatures exhibited the highest interactions with a lag of around a month, while air temperatures showed the
highest interaction at a lag of 26 days. Moisture-related variables all showed similar time lags, with relative humidity having
the highest interaction with NEE at 16-18 days prior, similar to SWCn and VPD at 18 and 20 days, respectively. EF had the
highest interaction with NEE at a lag of two weeks. Shortwave radiation-related variables showed a strongly lagged effect (i.e.,
PAR 59 days, SWDR 57 days), while the effect of longwave radiation LWDR is less lagged, showing the strongest interaction
26 days prior (Fig.3 (f)).
At NPT, both $r$ and $MI_{sync}$ detected soil temperatures, air temperatures and VPD as the most important NEE controls behind
gcc_gr and NDVI, with only minor differences. LWDR was more important using $r$ compared to $MI_{sync}$. SWCn and Rh were
in the middle ranks, PAR and SWDR in the lower ranks (Fig.3 (g), (h)). Additionally, NDVI and gcc_gr led NEE with the
strongest interaction at lags of 2 weeks and 10 days, respectively, followed by soil temperatures and air temperatures with the
highest interaction at a lag of around a month (Fig.3 (i)). EF showed the highest interaction at a lag of 12 days. Other moisture-
related variables like VPD, SWCn, and Rh were also detected to be in the middle ranks by $MI_{max}$, with time lags of 26, 23, and
20 days, respectively. PAR and SWDR showed the highest interaction with NEE at time lags of 53 and 52 days, respectively
(Fig.3 (i)).
MI and $r$ agreed in the detection of the most important drivers, thereby proving that information theory is applicable to our
case. Therefore, in the remainder of this paper we focus on values obtained using MI, as MI is able to detect collinear
relationships as well as leading and lagging effects. Additionally, we discuss variables with $MI_{max} > 0.2$ in the following
sections to concentrate on the information provided by variables with greater explanatory value.

**Figure 3: Pearson correlation coefficient (*r*) (a), (d), (g), synchronous mutual information (MI$_{sync}$) (b), (e), (h) and maximum mutual information within a 60-day window (MI$_{max}$) (c), (f), (i) between net ecosystem exchange (NEE) and potential drivers over the 7-year period (2016-2022) at the control plot CT (a)-(c), the nitrogen fertilized plot NT (d)-(f) and the nitrogen and phosphorus fertilized plot NPT (g)-(i). The color scale in the MI$_{max}$ plots indicates the day when MI$_{max}$ occurs, positive values indicate that the variable leads NEE, negative values vice versa.**





**3.3 Effect of Fertilization on NEE Sensitivity to its Controls**

The relationships between NEE and biogenic and environmental variables were asynchronous, as indicated by $MI_{sync} < MI_{max}$ for all variables. Therefore, we focused on $MI_{max}$ to describe the differences in NEE sensitivity to various controls across towers.

At NT and NPT, NDVI showed a slightly higher interaction with NEE (0.41 and 0.40) than at CT (0.36). However, the time lag was smallest at NT (12 days) compared to 15 and 16 days at NPT and CT, respectively. Thus, N fertilization appeared to shorten the reaction time of NEE to changes in NDVI. GCC at the grass level showed higher explanatory value for NEE at NPT and NT ($MI_{max} = 0.37$) compared to CT ($MI_{max} = 0.33$). EF showed only slight differences in interaction strengths among the sites. At CT, $MI_{max}$ was 0.29 with a lag of 15 days, $MI_{max}$ were 0.30 with lag times of 14 and 12 days at both NT and NPT, respectively (Fig. 4, S3).

Relative humidity at two heights showed the lowest interaction with NEE at CT with $MI_{max}$ values around 0.24, while the fertilized sites had slightly higher values of 0.26-0.27. Furthermore, the reaction time of NEE to relative humidity decreased with fertilization. At CT, the time lag was 24-27 days, at NPT it was 20-22 days and at NT 16-18 days. VPD showed the highest explanatory value for NEE at NT ($MI_{max} = 0.31$), followed by NPT ($MI_{max} = 0.30$) and CT ($MI_{max} = 0.27$). The interaction between NEE and air temperatures was slightly higher at the fertilized plots compared to the control. The $MI_{max}$ value was around 0.28 at CT, but higher at around 0.32 at NT and NPT. Soil temperatures showed similar interaction strength with NEE across treatments, with $MI_{max}$ ranging from 0.31 to 0.33 (Fig. 4, S3).

Regarding radiation variables, PAR showed slightly higher interaction with NEE at NT ($MI_{max} = 0.28$), than at NPT ($MI_{max} = 0.26$) and CT ($MI_{max} = 0.25$). Similarly, SWDR showed highest interaction with NEE at NT ($MI_{max} = 0.26$), while at NPT and CT it was equally strong ($MI_{max} = 0.24$).

In terms of soil variables, soil temperatures exhibited the strongest interaction with NEE. While soil temperatures below the canopy ($T_{soil}Shd$) were almost the same across sites ($MI_{max} = 0.33$), the importance of soil temperatures under open air differed at CT ($MImax = 0.31$) compared to the fertilized plots (0.34 and 0.33 at NT and NPT, respectively). SWCn showed the highest explanatory value for NEE at NT ($MI_{max} = 0.31$), followed by CT ($MI_{max} = 0.28$) and NPT ($MI_{max} = 0.27$) (Fig. 4, S3).

An overview plot with all variables including the ones with $MI_{max} < 0.2$ is provided in the Supplementary Material (S4).

Nutrient addition did not show a substantial effect on the sensitivity of ecosystem NEE to different drivers over the 7-year scale when considering the whole time series together. In the next step we examined the different seasons in greater detail.





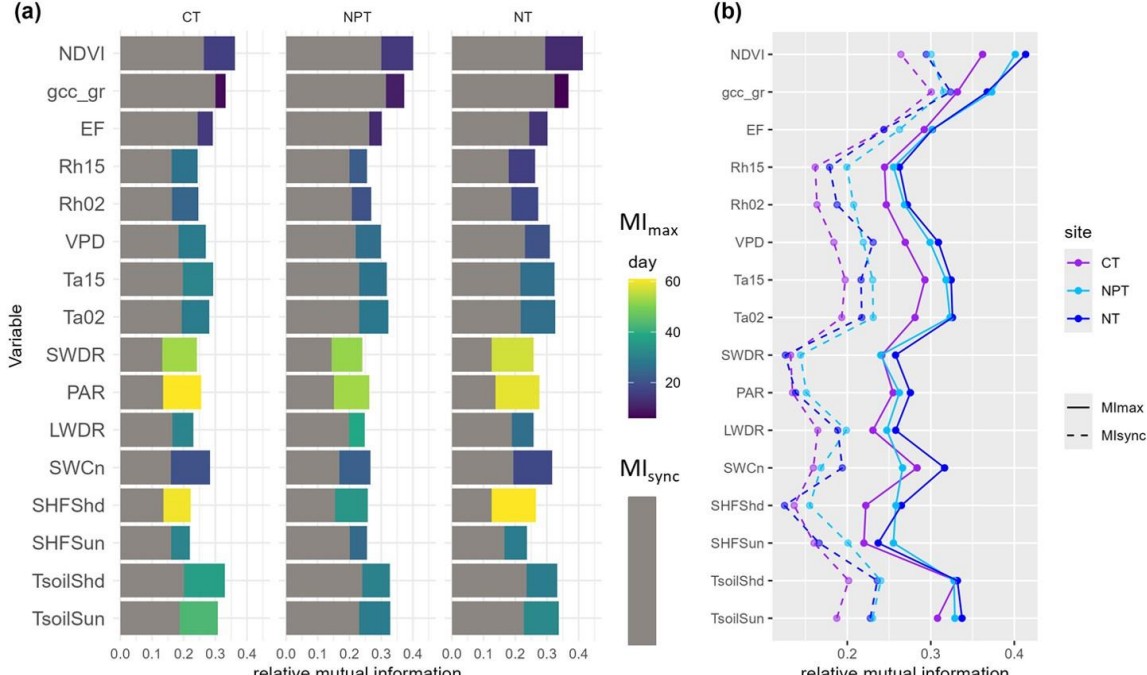

**Figure 4: (a) Synchronous ($MI_{sync}$, grey) and maximum ($MI_{max}$, colours) mutual information at the control site (CT), the nitrogen fertilized site (NT) and the nitrogen and phosphorus fertilized site (NPT) at the seasonal scale. The colour scale indicates the day when $MI_{max}$ occurs, with positive values indicating that the variable leads net ecosystem exchange (NEE) and vice versa. (b) $MI_{sync}$ (dotted lines) and $MI_{max}$ (solid lines) values at the three sites. Variables with $MI_{max} < 0.2$ are not shown here.**

### 3.4 Identifying Driver Importance in Different Phenological Seasons

We split the 7-year dataset into five different phenological seasons derived from grass layer GCC derived from PhenoCam photos, and calculated $MI_{sync}$ between NEE and each of the drivers. This analysis showed that the most important drivers differed between seasons and treatments (Table 2).

In the winter, the water vapor transfers of available energy, represented by EF, show a strong interaction with NEE at NPT and CT. Further, NDVI and tree-layer albedo, as well as radiation parameters such as PAR and SWDR were important in explaining NEE variations.

In the spring (i.e., the main growing season), NDVI and GCC at grass (gcc_gr) and tree (gcc_tr) levels showed the strongest interactions with NEE, indicating that NEE was dominated by photosynthetic activity (GPP) during this season. Furthermore, soil temperatures showed strong interactions with NEE at CT and NPT, but not at NT.

During the drydown phase, NEE was dominated by NDVI across treatments, with GCC at the grass level (gcc_gr) also showing strong interactions with NEE at CT and NPT. At NT, VPD exhibited a strong link with NEE, which was not as dominant at the other plots.



In the summer, soil temperatures showed high interactions with NEE, possibly relating to soil respiration. Additionally,
radiation parameters (i.e., SWDR, PAR) were important in explaining NEE variations during this season. At CT and NT, tree-
layer GCC became important, which was logical as the grass layer becomes senescent in the summer and is dormant in terms
of the ecosystem carbon flux. At NPT, gcc_gr showed a higher interaction with ecosystem NEE than gcc_tr.
In autumn, the regreening starts with the onset of rains, and NDVI and grass layer GCC (gcc_gr) showed strong interactions
with NEE, as GPP starts to dominate NEE again, driven by photosynthetic activity. Additionally, soil temperatures had a strong
link with NEE (as soil respiration is also high in this season), strongest at CT, as well as air temperatures.

**Table 2: Five most important drivers in each phenological season at each tower derived using synchronous mutual information. CT**
**= control site, NT = nitrogen fertilized site, NPT = nitrogen + phosphorus fertilized site.**

|  |  | CT | NT | NPT |
|---|---|---|---|---|
| **Winter**<br>wet and energy limited | 1. | Alb_tr | PAR | EF |
|  | 2. | PAR | Alb_tr | NDVI |
|  | 3. | EF | SWDR | PAR |
|  | 4. | $T_{soil}Shd$ | NDVI | SWDR |
|  | 5. | $T_{soil}Sun$ | SHF_Sun | $T_{soil}Sun$ |
| **Spring**<br>main growing season | 1. | gcc_gr | NDVI | NDVI |
|  | 2. | NDVI | gcc_tr | gcc_gr |
|  | 3. | gcc_tr | gcc_gr | gcc_tr |
|  | 4. | $T_{soil}Sun$ | Alb_tr | $T_{soil}Sun$ |
|  | 5. | $T_{soil}Shd$ | EF | $T_{soil}Shd$ |
| **Drydown**<br>senescence of grass layer | 1. | NDVI | NDVI | NDVI |
|  | 2. | $T_{soil}Shd$ | $T_{soil}Sun$ | gcc_gr |
|  | 3. | gcc_gr | $T_{soil}Shd$ | $T_{soil}Shd$ |
|  | 4. | EF | VPD | $T_{soil}Sun$ |
|  | 5. | $T_{soil}Sun$ | EF | EF |
| **Summer**<br>Dormant/dead grass layer | 1. | PAR | SHF_Shd | $T_{soil}Sun$ |
|  | 2. | $T_{soil}Sun$ | PAR | $T_{soil}Shd$ |
|  | 3. | SHF_Shd | SWDR | SHF_Sun |
|  | 4. | gcc_tr | gcc_tr | gcc_gr |
|  | 5. | SHF_Sun | NDVI | PAR |
| **Autumn**<br>Regreening of grass layer with onset of rains | 1. | $T_{soil}Shd$ | NDVI | NDVI |
|  | 2. | gcc_gr | gcc_gr | gcc_gr |
|  | 3. | NDVI | $T_{soil}Shd$ | $T_{soil}Shd$ |
|  | 4. | Ta15 | Ta02 | $T_{soil}Sun$ |
|  | 5. | $T_{soil}Sun$ | Ta15 | Ta15 |





## 3.5 Changes in NEE Sensitivity over Time

We observed that with N addition, NEE became less sensitive to certain variables during autumn (i.e., the regreening phase), the drydown phase, and winter over time (Fig.5). Specifically, in autumn, the sensitivity of ecosystem NEE to changes in air temperature (Ta15), shortwave radiation (SWDR and PAR), and NDVI decreased significantly over the 7-year period. In the drydown phase, the sensitivity of ecosystem NEE to changes in relative humidity (Rh02 and Rh15) and soil heat flux (SHF_Sun) also decreased significantly. In winter, however, we observed a significant increase in the sensitivity of NEE to variations in PAR, SWDR, and grass layer GCC (gcc_gr).

With the addition of N+P, significant changes in NEE sensitivity over time were observed in all seasons except the drydown phase (Fig.5). In autumn, the fertilization with N and P led to a significant decrease in NEE sensitivity to air and soil temperatures (Ta02 and $T_{soil}$Sun), PAR, and VPD. In spring, which is the main growing season, NEE sensitivity to variations in PAR increased significantly over time. In summer, NEE became significantly more sensitive to changes in grass layer GCC. In winter, NEE shows a significant increase in sensitivity to changes in both grass layer GCC (gcc_gr) and soil water content (SWCn).

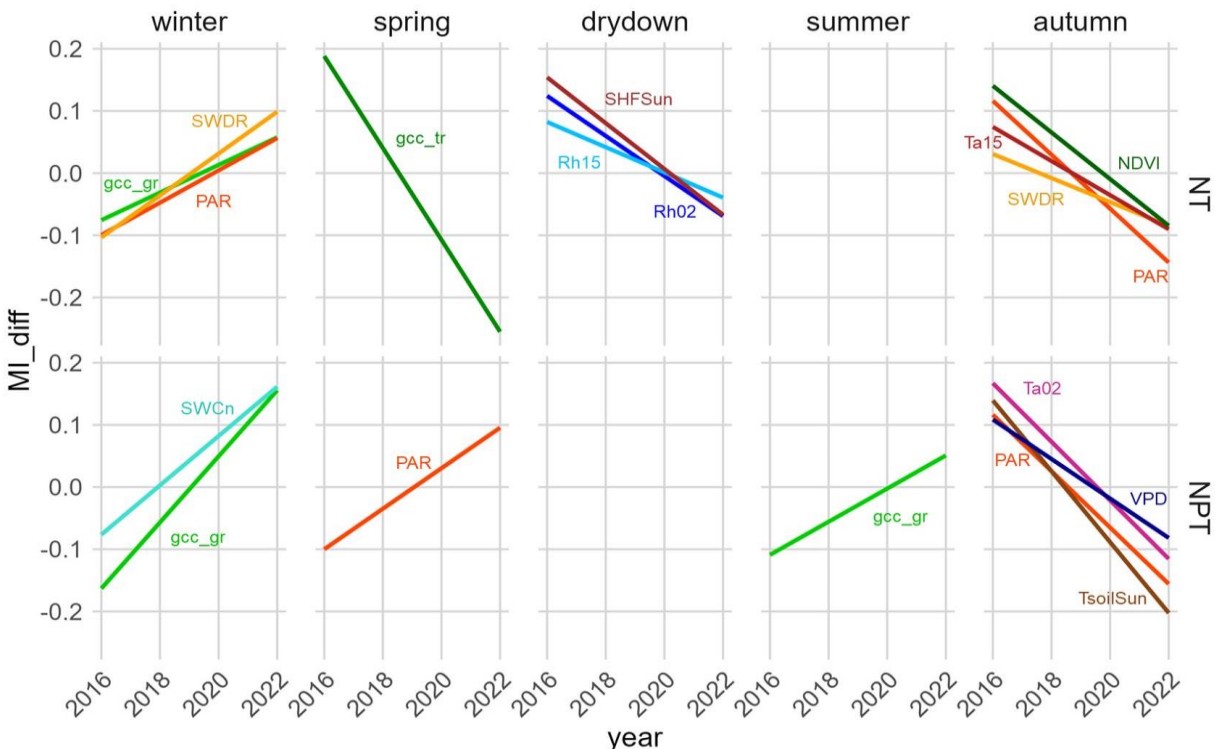

467

**Figure 5: Linear regressions of the seasonal synchronous mutual information difference (MI$_{diff}$) between NT and CT (bottom) and between NPT and CT (top) in different phenological seasons. Only the relationships with significant trends are shown. Significance level is set at $p < 0.05$. Variables with overall MI < 0.2 at all towers are not shown here.**



**4 Discussion**

**4.1 Nutrient addition increases seasonal NEE variability, driven by grass layer**

Our results indicate that nutrient addition enhances seasonal NEE variability compared to the control. Additionally, the seasonal variability increases over time at the fertilized plots. Looking at the difference between annual NEE maximum and annual NEE minimum, we notice a significantly increasing trend at the NPT plot (Fig.6). We argue that this nutrient effect is dominated by grass layer which substantially controls the NEE dynamics in this system. Our analysis supports that grass layer GCC and NDVI are most important in explaining NEE variations across treatments (Fig. 3, 4). Both variables represent grass layer greenness, as the larger fraction of the surface consists of annual grasses (Bogdanovich et al., 2021) and remotely sensed NDVI is dominated by the herbaceous layer.

The added nutrients mostly stay in the herbaceous layer at the study sites (El-Madany et al., 2021), and it is therefore more affected by the nutrient manipulation than trees. It has been found that the nutrient addition leads to higher root biomass and root length density (Nair et al., 2019) and N can be absorbed and used for leaves. In the leaves, N enhances the photosynthetic capacity (Fleischer et al., 2013) which supports the faster increase in maximum GPP and biomass in the fertilized plots, as confirmed by Luo et al. (2020). NT and NPT show higher productivity and therefore higher biomass amount compared to the control (Luo et al., 2020). As the grass layer is senescent in summer, this results in a higher amount of dead biomass, which will then be respired by soil microbes (Manzoni et al., 2020; Moyano et al., 2013) as soon as there is sufficient water available (Huxman et al., 2004). This indicates that there is a higher carbon turnover at the fertilized plots, leading to an increased range of NEE within a year (Fig.6). It agrees with findings from Ma et al. (2016), who found in a Californian oak grass savanna that the amount of grass litter determines the size of the fast carbon pool in consecutive seasons.

Evergreen tree species have relatively constant foliage amount throughout the year and are able to use their deeper roots to access lower water resources in the soil (Baldocchi et al., 2004; Rolo and Moreno, 2012), the herbaceous layer is strongly dependent on rainfall variations as it accesses water in the topsoil with a dense near-surface root system (Ward et al., 2013). It is therefore much more sensitive to intra- and inter-annual climate variations (Luo et al., 2020). This is probably the reason that the seasonal NEE variability was very low at all sites in 2017. We attribute this to extraordinary dryness in that year, as dryness can lead to severe decreases in both GPP and $R_{eco}$ in this type of ecosystem (Ma et al., 2007).





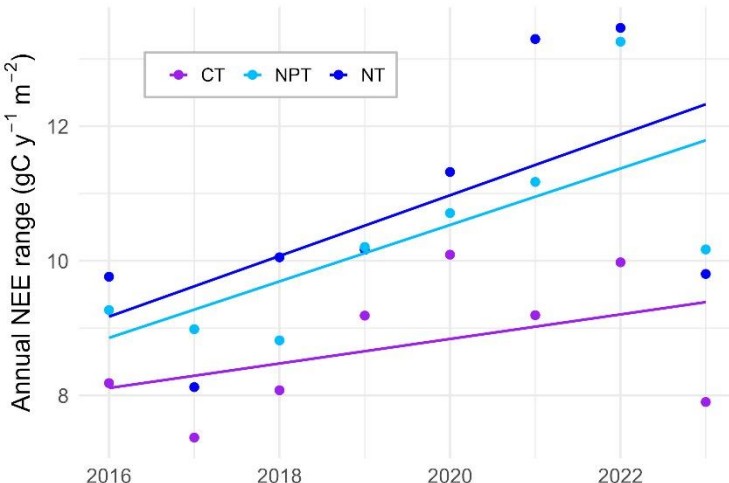

498

**Figure 6: Annual range of net ecosystem exchange (NEE) (i.e., maximum NEE minus minimum NEE) in gC m⁻² y⁻¹ calculated using the seasonal reconstructed signal at the control site (CT), the nitrogen fertilized site (NT) and nitrogen + phosphorus fertilized site (NPT). The range at NPT (p-value = 0.049) significantly increased over 8 years, while not at NT (p-value = 0.116) and CT (p-value = 0.270).**

503

## 4.2 Key controls of seasonal NEE

Our results indicate that proxies for vegetation greenness (NDVI and GCC at grass layer derived from satellite and PhenoCam data, respectively) are the primary factors influencing the seasonal NEE signal in this ecosystem across treatments (Fig.3). However, depending on different seasons, other variables such as air temperatures, VPD, moisture-related variables, and soil temperatures can also be important.

Many studies identify NDVI, a proxy for vegetation greenness and photosynthesis, as a primary predictor of NEE (Del Grosso et al., 2018; Hermance et al., 2015; Morgan et al., 2016). NDVI, generally derived from satellite data, represents ecosystem greenness and its connection with ecosystem $CO_2$ fluxes has been intensively studied (Barnes et al., 2016; Hermance et al., 2015; Morgan et al., 2016; Running and Nemani, 1988). However, quantifying the importance of coexisting vegetation layers is more complex and less understood. Digital repeat cameras and vegetation greenness indices derived from them provide a powerful tool for analysing the greenness of different plant types (Migliavacca et al., 2011; Petach et al., 2014; Richardson et al., 2009; Yan et al., 2019) and their influence on ecosystem fluxes (Luo et al., 2018; Moore et al., 2017; Wingate et al., 2015). Our analysis confirms that grass layer dynamics are dominant in controlling seasonal ecosystem NEE at this site.

In situ measurements of vegetation greenness, however, are not available at all EC sites. We found that EF (i.e., evaporative fraction), representing the fraction of available energy transported by LE, is the third most important driver across treatments and methods on the synchronous scale. EF is strongly influenced by net radiation and water-related variables like soil moisture and VPD (Gentine et al., 2007; Tong et al., 2022). Since it depends on the portion of LE that is transpired by plants, it is also





impacted by LAI (Gentine et al., 2007). EF therefore serves as a bridge between meteorological and vegetation controls. We
suggest that at semi-arid sites where GCC measurements are not available, EF, calculated from measured LE and H, can serve
as important predictor of NEE.
In water-limited semi-arid ecosystems, NEE variations are typically dominated by soil-water related variables such as SWCn
and precipitation (Archibald et al., 2009; Baldocchi and Arias Ortiz, 2024; Huang et al., 2016b; Morgan et al., 2016). These
variables usually exert a greater influence than radiation and temperature (Del Grosso et al., 2018; Kannenberg et al., 2024).
Water availability promotes plant photosynthesis (Parton et al., 2012), but rain pulses can also enhance heterotrophic
respiration rates (Morgan et al., 2016). While we do not use precipitation data for the MI analysis, as it tends to be zero on
many days and cannot be used in MI (Gong et al., 2014), SWCn can capture topsoil moisture and indicate precipitation pulses.
Additionally, EF can serve as a proxy for these pulses. In our analysis we identify EF as one of the most important NEE drivers,
while other moisture-related variables (e.g., SWCn, VPD, Rh) are generally ranked lower in importance compared to air and
soil temperatures (Fig.3).
Air temperature can directly affect the speed of the enzyme responsible for carbon fixation and the rate of photosynthetic
electron transport (Leuning, 2002; Xu and Baldocchi, 2003). Additionally, temperature impacts the availability of
photosynthetic enzymes, membrane fluidity, and the expression of associated proteins (Yamori et al., 2014). However, our
results show that soil temperatures, both under oak trees and in open areas, play a significant role in explaining seasonal
ecosystem NEE variations (Fig.3), exceeding the importance of air temperatures. Soil respiration, one of the components in
$R_{eco}$, is highly sensitive to soil temperature (Conant et al., 2000), and elevated soil temperatures are associated with increased
soil respiration in semi-arid ecosystems (Richardson et al., 2012). These temperatures influence heterotrophic respiration,
which constitutes a substantial part of ecosystem NEE at our site (Casals et al., 2011). The high importance of soil temperatures
hints to $R_{eco}$ dominating ecosystem processes and is especially relevant as the trend of increasing soil temperatures is stronger
than increasing air temperatures in the Mediterranean, particularly in grasslands with low soil moisture availability (Wang et
al., 2024).
Radiation parameters, in particular PAR, do not appear to play a crucial role on the seasonal scale. While other studies have
identified it as a major control of NEE in semi-arid ecosystems (Baldocchi and Arias Ortiz, 2024), we argue that PAR
predominantly influences the daily NEE signal (S1.2), but its importance diminishes on seasonal time scales.
Overall, we observe only marginal differences between the treatments when considering the 7-year period (2016-2022)
together. The added nutrients, particularly N, are primarily absorbed by the herbaceous layer (El-Madany et al., 2021) that
senesces annually. Consequently, some of the added nutrients may be lost from the system, diminishing the long-term effect
of the fertilization. By calculating $MI_{sync}$ and $MI_{max}$ for one year post-fertilization (March 2016-February 2017), we observe
greater differences between the three plots in MI values and lag times (S5). Additionally, the ecosystem is strongly water-
limited in the summer and energy-limited in the winter (Luo et al., 2018; Nair et al., 2019). These limitations can be more
pronounced than nutrient limitations in their respective seasons, overshadowing the effects of added nutrients when analysing





the entire dataset together. Therefore, we divided the dataset into five phenological seasons to gain deeper insights into how
added nutrients and altered stoichiometric balance affect seasonal NEE.

**4.3 Fertilization effects in different phenological seasons**

Looking into phenological seasons gives a deeper insight into how environmental variables influence seasonal NEE and how
N:P levels affect this relationship. We find that nutrient addition has an effect on NEE - control relationships when other
limitations are not too strong.
In the primary growing season, spring, NEE is dominated by GPP. The key drivers during this season across sites are NDVI
and GCC of both the herbaceous and tree layers (Table 2). Water is typically abundant promoting plant photosynthesis during
moderate temperatures in this time (Baldocchi and Arias Ortiz, 2024). These conditions are further supported by increased day
length and higher radiation levels (Luo et al., 2018). The rise in incoming radiation, extended daylight hours, and elevated
temperatures, coupled with the increased atmospheric evaporative demand (i.e., higher VPD), lead to a strong correlation
between precipitation and both GCC and GPP, as observed in various Mediterranean ecosystems (Diodato and Bellocchi, 2008;
Luo et al., 2018; Ma et al., 2007).
During the regreening of the herbaceous layer starting in autumn, NDVI shows the strongest interaction with NEE at the
fertilized plots - but not at the control plot. This aligns with previous studies showing that the green-up in this season happens
faster and the maximum GPP is higher at the fertilized plots, resulting from larger resource utilization at NT or improved
resource use efficiency at NPT (Luo et al., 2020). With the increase in soil moisture in early autumn, a greater quantity of
organic and inorganic nutrients becomes available to plants (Agehara and Warncke, 2005; Luo et al., 2020). N availability in
the soil is expected to be highest in this time (Morris et al., 2019), leading to higher net carbon uptake rates (El-Madany et al.,
2021). Leaves quickly expand and pigments rapidly increase during this green-up period (Croft et al., 2015). At CT, the green-
up happens later compared to the fertilized plots and NEE is dominated for a longer time by $R_{eco}$ instead of photosynthetic
activity (Luo et al., 2020). Our results indicate that soil temperatures below oak trees are more important than those in open
areas during this season (Table 2). The carbon pools under oak trees are the largest, providing substantial material for
heterotrophic decomposition (Casals et al., 2009). During autumn, after a prolonged dry season where a significant amount of
litter and organic material has already been decomposed by microbes, litter remains available for further heterotrophic
decomposition mainly below the trees. This ongoing decomposition under oak trees contributes to $R_{eco}$, especially as the onset
of rains enhances microbial activity due to increased water availability (Borken and Matzner, 2009). Additionally, the topsoil
layer remains wet for longer after rain pulses under oak trees compared to open areas, as soil moisture is primarily influenced
by soil evaporation in this season as the soil is rather bare. Therefore, differences in soil respiration between open and shaded
pastures can also be attributed to variations soil moisture.
In winter, the ecosystem is energy-limited (Luo et al., 2018), therefore radiation components (i.e., PAR and SWDR) are
important predictors for NEE. Tree Albedo shows strong interactions with NEE at CT and NT, and NDVI shows strong
interactions with NEE at both fertilized plots. Plant growth is enhanced by added nutrients (Luo et al., 2020) and made available





by abundant water availability (Lee et al., 2010) in this season. Also, N+P addition can lead to an increased species diversity
due to alleviated nutrient limitation facilitating the co-existence of multiple species (Köbel et al., 2024). Additionally, EF
shares high mutual information with NEE variations. This is likely because respiration does not change significantly during
this period, and VPD is relatively low, leading to a strong coupling between NEE and LE. Additionally, in winter, the stomatal
control of the tree transpiration is not too strong, as soil water is abundant (Klein et al., 2013).
In the water-limited seasons, the nutrient effect is minimal as the grass layer is dormant and nutrients are not made available
due to a lack of water. During the drydown period, soil moisture (i.e., SWCn) decreases drastically due to increasing air
temperatures and scarce rainfall (Battista et al., 2018; Luo et al., 2018). This induces annual grasses to become senescent,
leading to a loss of chlorophyll content (Luo et al., 2018). The rate of this senescence can determine whether NEE becomes
positive or negative during this time. NDVI and grass layer GCC, the most important predictors of NEE in this season across
sites, can provide insights into the dry down rate. At NT grass layer GCC is less important, which we attribute to a more rapid
drydown, causing the grass layer to enter dormancy earlier than at other sites (Luo et al., 2020). This is because N addition
promotes faster water usage (Luo et al., 2020), accelerating the decrease in SWCn and thereby hampering photosynthesis. It
leads to a higher transpiration at NT compared to the other sites, potentially due to rhizosphere priming to increase P
mobilization through microbes, as adding only N to the system leads to a P deficiency (El-Madany et al., 2021). In addition,
N fertilization can alter species diversity and composition, likely selecting for species that senesce early (Wang and Tang,
2019). The higher interaction of soil temperatures with NEE in this season compared to the wetter seasons, show that $R_{eco}$
starts dominating NEE, as $R_{eco}$ is strongly connected to soil temperatures (Metz et al., 2023). VPD is a stronger control of NEE
at NT compared to the other two plots. Transpiration is highest at NT, as plants transpire more to obtain limited P from the
soil (El-Madany et al., 2021; Pang et al., 2018; Rose et al., 2018). It is therefore more sensitive to changes in VPD.
In summer, the driest period at the ecosystem, $R_{eco}$ dominates NEE and thus we find a strong interaction between NEE and
soil temperature and soil heat flux (i.e., SHF_Sun and SHF_Shd). Besides, PAR is important for predicting seasonal NEE,
showing the strongest interaction at CT. The importance of PAR is lower at NT and lowest at NPT. N+P addition increases
the light use efficiency most because P has a positive effect on photochemical quenching in leaves and on active fluorescence
measurements (Martini et al., 2019; Singh and Reddy, 2014), leading to less dependency of NEE to radiation parameters at
that site. At CT and NT, tree layer GCC is important as the grass layer becomes senescent in the summer and is dormant in
terms of ecosystem carbon flux. Since the greenness of the oak trees is constant throughout the year, GPP is mainly determined
by the tree layer in the summer months (Luo et al., 2018). However, gcc_gr shows a higher interaction with NEE than gcc_tr
at NPT. Even though most of the grass layer is mostly dead in this season, there are some perennial species (e.g. *cynodon*
*dactylon*) remaining green for longer in summer and can regreen after any rain events (personal communication with local
collaborators). Therefore, N+P addition very likely leads to a consequential change in species composition (Köbel et al., 2024)
with an increase in these perennial species or results in an increase in their productivity. So far it has been found that N+P
addition can lead to an increasing number of forbs (Köbel et al., 2024), which tend to senesce later than other herbaceous



species at the site (Luo et al., 2020). Nevertheless, the occurrence of summer-green species following nutrient addition will
have to be investigated further.
The analysis of driver importance in different phenological seasons provides significant insights into ecosystem processes.
However, some variables must be interpreted with caution. The soil properties at this site are highly heterogeneous, which
affects the representativity of variables like soil temperature, soil water content and soil heat flux in the EC flux footprint (Luo
et al., 2018; Paulus et al., 2022). This is particularly relevant given the substantial differences between below-canopy and
open-air soil conditions. To address this, we have separated the measurements into areas under the oak tree canopy and sunlit
areas ($T_{soil}$Shd and SHF_Shd, and $T_{soil}$Sun and SHF_Sun). Despite this effort, the local soil heterogeneity is more complex,
influenced by varying proportions of sand, clay, and soil organic carbon (Casals et al., 2011; Weiner et al., 2018). Therefore,
it is important to consider that these measures may not fully capture the sensitivity differences in the ecosystem.

**4.4 Future implications**

In winter, the ecosystem has abundant water availability, and energy becomes the primary limiting factor after nutrients were
added. With N only addition, we observe that NEE becomes significantly more sensitive to changes in the radiation
components, PAR and SWDR (Fig.5). However, the addition of N+P results in a significant increase in sensitivity to changes
in soil water content rather than radiation components. N+P addition enhances water use efficiency in the ecosystem (El-
Madany et al., 2021; Martini et al., 2019), and consequently, water can be used more efficiently for photosynthesis with
similarly low radiation levels and increased water availability could lead to a higher GPP. N fertilization primarily affects the
herbaceous layer (El-Madany et al., 2021), and our results agree with this, showing a significantly increased sensitivity of NEE
to grass layer greenness in winter at NT and an even steeper increase at NPT (Fig.5). At N+P plot there are more nutrients
available at a higher N:P stoichiometric balance.
In spring, the sensitivity to tree layer greenness decreases with N fertilization. An experimental study by Biro et al. (2024)
supports these findings, demonstrating that N addition results in decreased tree growth due to competition with grass, which
also intensively forages for P. The study suggests that grasses likely prevail in below-ground competition, primarily due to
their substantial root biomass allocation and investment in nutrient-acquiring enzymes, such as phosphatase. These adaptations
enable grasses to efficiently sequester both N and P from the soil, thereby outcompeting trees for these essential nutrients (Biro
et al., 2024; Rolo and Moreno, 2012).With the addition of N+P, we observe that the NEE sensitivity to PAR increases
significantly in spring (Fig.5). Water and nutrients are abundant in this season at NPT, making the availability of energy more
crucial.
In the water-limited seasons, ecosystem processes behave quite differently and we observe less effect of nutrient addition.
With N+P addition, there is no significant trend in NEE sensitivity to its drivers, except for a significantly increased sensitivity
to grass layer greenness in summer. This agrees with our previous findings that in summer gcc_gr is amongst the most
important drivers at NPT (Table 2). This reflects changes in the species decomposition with N+P fertilization, enhancing



especially the growth and diversity of forbs and perennial species. We argue that long-term N+P addition could even lead to
an increased productivity, leading to an increasing importance of grass layer greenness for ecosystem NEE.
The significant increase of the yearly NEE range at NPT over time (Fig.6) is very likely caused by the increased NEE sensitivity
to drivers in spring and summer, as the minimum NEE (usually occurring in spring) becoming more negative and maximum
NEE (usually occurring in summer) becoming more positive. Consequently, the increased NEE sensitivity to changes in PAR
in spring and increased sensitivity to gcc_gr in summer might enhance the size of this annual range.
In the drydown phase we observe that with N addition, the sensitivity of ecosystem NEE to changes in relative humidity (i.e.,
Rh02 and Rh15) and SHF_Sun decreases significantly. This indicates that the ecosystem might become more resistant against
variations in these variables in the future.
In autumn both fertilized sites become less sensitive to changes in atmospheric variables such as the radiation components
PAR and SWDR, air temperatures and VPD, compared to the control plot. This indicates that water availability is
predominantly important for NEE with added nutrients, and the sensitivity to the other variables decreases. It is possible that
either the vegetations or the microbes become less restricted by these variables.
We conclude that with more N-input from human activities entering terrestrial ecosystems (Penuelas et al., 2013), savannas
may become less sensitive to environmental factors like humidity, radiation, and temperature during the transitional seasons
(i.e., drydown and regreening). These seasons determine the start and end of an active grass layer and therefore dominate the
annual carbon balance of the ecosystem. In addition, we expect the NEE variability to increase even more in the future with
more N deposition and a changing climate. To note, the results of this study cannot explain on how the long-term nutrient
addition affects the ecosystems resistance to extreme events.

## 5    Conclusion

We analysed a long-term (2016-2022/23) dataset of flux, biometeorological, satellite and PhenoCam data from the semi-arid
experimental site, Majadas de Tiétar, to evaluate the importance of different drivers for NEE across three different nutrient
balances. To detect the most important drivers, we used only daytime daily values of observed data to extract the seasonal
signal of all variables using the Singular Spectrum Analysis.
With both Pearson correlation and mutual information analysis we show that the grass layer drives seasonal variations in NEE
across all treatments, and that both N and N+P addition increases the seasonal NEE variability. We find that soil temperatures
are more important in explaining NEE variations than previously expected. When looking into the entire 7-year data together,
the water and energy limitation cycles overshadows the nutrient addition effect. Dividing the dataset into phenological seasons
reveals how environmental variables and nutrient manipulation influenced NEE on a seasonal scale. Altered nutrient levels
affect NEE-control relationships when water and energy limitations are not too strong, particularly during the primary growing
season in spring, where NDVI and grass layer GCC are key drivers. In autumn, NDVI shows the strongest interaction with



NEE at fertilized plots, indicating faster green-up and higher GPP due to enhanced nutrient availability. During drier seasons, nutrient effects are less pronounced as the grass layer becomes dormant.

N and N+P additions significantly alters the sensitivity of NEE to environmental controls over time. In winter, N addition increases NEE sensitivity to radiation, while N+P addition increases its sensitivity to changes in soil water content. In spring, N+P addition increases sensitivity to PAR. The herbaceous layer primarily benefits from nutrient additions, leading to increased sensitivity of NEE to grass layer greenness and decreased sensitivity to tree layer greenness. During water-limited seasons, nutrient effects were minimal, except for increased importance of grass layer GCC in summer at NPT, indicating an increase in abundance and/or productivity with N+P treatment due to changed species composition and higher biodiversity. We conclude that with increasing anthropogenic N deposition the carbon dynamics of savannas might become even more variable in the future, but more resistant to variations in some atmospheric variables in the transitional seasons, which determine the annual carbon balance of the ecosystem. However, their responses to extreme events in the future remains to be explored.

**Data Availability**

Ecosystem level data is available on the European Fluxes Database (https://www.europe-fluxdata.eu/) and PhenoCam (https://phenocam.nau.edu/webcam/). The FluxnetEO dataset can be found on the ICOS Carbon Portal.

**Author Contributions**

LN: Conceptualization, Data curation, Methodology, Formal analysis, Writing – original draft preparation. TE: Data curation, Writing - review & editing. JN: Writing – review & editing. AC: Data curation, Writing – review & editing. GM: Data curation, Writing – review & editing. RN: Software, Writing – review & editing. YL: Software, Writing – review & editing. AH: Supervision, Writing – review & editing. VR: Data curation. Writing - review & editing. MR: Resources, Funding acquisition, Writing – review & editing. SL: Conceptualization, Methodology, Supervision, Writing – review & editing.

**Declaration of Competing Interest**

The authors declare that they have no known competing financial interests or personal relationships that could have appeared to influence the work reported in this paper.



**Acknowledgements**

We thank Mirco Migliavacca (European Commission, Joint Research Centre, Ispra, Varese, Italy) for sharing advice and expertise, the Freiland Group at Max-Planck Institute for Biogeochemistry in Jena for maintaining and calibrating sites and sensors, as well as Ramón López for daily data collection and site maintenance.

The nutrient manipulation experiment was funded by the Alexander von Humboldt Foundation Max Planck Research Prize to Markus Reichstein (i.e., MANIP project). Laura Nadolski received financial support from the International Max Planck Research School for Biogeochemical Cycles (IMPRS-gBGC).

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
