# Peer review of "Altered Seasonal Sensitivity of Net Ecosystem Exchange to Controls Driven by Nutrient Balances in a Semi-arid Savanna"

_EGUsphere, 2024_

## Referee Comment (RC1)

In a world dealing with a changing climate, there is a need for studies investigating environmental changes following anthropogenic influences, especially in understudied ecosystems with complex dynamics such as semi-arid savannas. This study uses an unique long term dataset collected in a large-scale nutrient addition experiment in a semi-arid savanna in Spain to look into the effect of altered nutrient levels on the relationships between NEE and it's key drivers, using robust methods as Singular Spectrum Analysis and Information Theory. The long term dataset is analyzed both as a whole and divided into phenological seasons, which results in a deeper understanding of the ecosystem as wel as interesting insights into the effects of the nutrient addition, underneath the water or energy limitation during different seasons. The methods are well explained and the important results are wel discussed, however some points require further clarification or discussion.

**Specific comments:**

*Materials and Methods*

1. Line 109: Authors could add a map of the region

2. Line 137: Is there more information on when N and P were applied to the plots in terms of seasons or years? Would you suspect that the results over the years, for example in Fig 6, could be in anyway linked to the timing of the application of N and P?

3. Line 154. How many soil sensors were installed per footprint? And were they installed in open field or under trees or both? There is mention in the discussion part that soil temperatures below oak trees are more important than those in open areas during the regreening in autumn (line 575) and that this also could be related to the variations in soil moisture between open and shaded pastures. Therefore it seems important to know where the sensors were located and if the authors were able to capture some of these variations in soil moisture that could underpin this statement.

4. The authors use only daytime measurements to calculate the aggregated daily means of NEE. Why is the nighttime data removed?

5. Line 228 – 230 "In the second step X is decomposed into its orthogonal components by determining eigenvalues and eigenvectors corresponding to principal components (singular value decomposition). Then the eigenvalues of the covariance matrix X · X are ranked."

   I don't know the SSA method very well, however this part is slightly confusing for me as I think it is the eigenvalues and eigenvectors of $XX^T$ that are determined and then X is decomposed in matrices with rank 1 which are constructed using these eigenvalues and eigenvectors. For me the term "decomposed in its orthogonal components" sounds vague. Also maybe avoid the use of "ranked" as I think you mean ordered by decreasing magnitude here. Rank in terms of matrices can be confused with the terminology of rank of a matrix.

6. Line 239: Here is stated that the data is gap-filled using the igapfill – function, however in Line 205 – 206 you state that you only use measured values to avoid confounding with the other meteorological variables in later analyses. Is this gap filling necessary because the

SSA requires a full time series? Will you not introduce this confounding again by gap filling here, in the sense that even though you remove the gap filles values again after the SSA (mentioned in Line 299), the gap filling will have an influence on the SSA result? Or is this method of gap filling not based on the meteorological data?

7. Section 2.4.4: I think that there are some inconsistencies with equations here. Formula (4) has no double sum, iterating over both x and y. This double sum is however seen in equation (6). Equation (6) on the other hand has no logarithm included in the right hand side of the equation both in the numerator as in de denominator (as is present in equation (4)) . Also a maximum operator should be present as $MI_{max}$ is the maximum iterated over different values of tau and in the denominator of equation (6) an "i" pops up. Maybe check that these formulas are indeed consistent and correct.

*Results*

8. Section 3.2.

   Line 351 – 354 states the common, most important predictors which are nicely highlighted and explained in the discussion, section 4.2

   Line 385 – 388 highlights that r and MI nicely agreed in the detection of the most important drivers, which explains why you continue only with the MI measurements.

   Line 355 – 384 describes the key controls for each plot, however I would suggest to make this shorter and more to the point. It reads difficult due to the many variable names and these separate results for each plot are nowhere discussed in the discussion section. The comparison between the towers in section 3.3 contributes more to the story of the paper than these separate observations, in my opinion.

   The section and graph are needed to support discussion section 4.2 but I feel like the separate observations could be either shortened or restructured.

9. Table 2: the authors might add the MI values in the table or in supplementary material of the five most important drivers for each phenological season. Would be interesting to see if for example the three first ones have a way higher MI value than the two last ones, or if the values are all close to each other.

*Discussion.*

10. Section 4.1
    Line 475: there is an increasing trend in the difference between annual NEE maximum and annual NEE minimum in the NPT plot. Is both the minimum value going down and the maximum value going up? Or is this increase in difference guided by mainly one of the two?

    Only NPT has a significant increase, however the explanation as why this could be the case, does not specifically mention only N+ P addition, so you would also expect this increase in the N plot. Is there an explanation why this is not the case? Or would you nuance the p-value and suggest that this increase might also be the case for N addition?

11. Line 530: here is stated that EF, which is a proxy for rain pulses, is an important driver but that SWC, VPD and RH are not important drivers following the MI analyses. This seems to be contradictory results as you mention both of them as proxies for rainfall. Points this towards the fact that in this study soil water related variables and rain pulses are not as important as previously found in other studies and that the link with EF as important driver is more based on its relation with LAI or radiation than with moisture related variables? However in line 544 is also stated that Radiation does not seems to have a major influence. How can this opposite results be interpreted?

12. Section 4.3: the authors could restructure the sections to follow the order of Table 2 and section 3.4. or restructure the table and the accompanying result section to follow the order from the discussion section.

**Technical corrections:**

*Abstract*

Line 15: Semi-arid ecosystems dominate the variability and trend of the terrestrial carbon sink.

Line 29 -30 : The increasing NEE variability might become even more pronounced with increasing N deposition and a changing climate in the future.

*Introduction*

Line 62:  man-made savanna-like agroecosystem, …

Line 72-73: Few studies so far have dealt  with …

*Materials and Methods*

Table 1: add in the caption that the soil heat flux and soil temperature in the shadow were calculated based on the shadow fraction …
Also the height of the air pressure device is not present

Line 113: heterogeneous with values between 0.5 and 2.5 ….

Line 151: here CO2- flux is also in the list of additional atmospheric variables, maybe remove

Line 173: and collected red, blue, ….

Line 291: Positive and negative values of $\tau$ show an asynchronous interaction between X and Y, with a lag …

This phrasing may sounds like it insinuates that it is the value of tau that is the interaction, however tau is merely the time lag. Maybe rephrase.

*Results*

Line 347-350:  this sentence is difficult to understand and has a point which seems wrongly placed.

"accounts for collinear relationships. and MIsync and ..."

Line 411: remove the sentence about soil temperature here, soil temperature is again mentioned in a better way in Line 416

Line 455: "We observed that with N addition, NEE became less sensitive to certain variables during autumn (i.e., the regreening phase), the drydown phase, and winter over time (Fig.5)."

In winter this is "more sensitive" and not "less sensitive" as there is increase as mentioned in line 459.

Figure 5: the caption states that NT vs CT is in the bottom figure and NPT vs CT in the top figure but it is the way around.

---

## Referee Comment (RC2)

This is a sound piece of work showcasing the use of eddy covariance measures of ecosystem fluxes as an experimental tool, with measures examining impacts of N and P additions to semi-arid savanna ecosystem. Tree-grass savannas occupy ~30% of the global land surface and quantifying fluxes from these systems is critical as they appear to be a source at present to the atmosphere and they drive global variability of global atmospheric CO2.  This is a unique experiment, and the EC work described here is a great add on, highlighting the power of long-term studies - another 7 or 8 years we will start to pick up impacts of CO2 fertilisation, heatwaves, increasing variability of precip, as well as the +N and P effects reported. Given the novel statistical analytics described in this study, untangling all of these interacting drivers of C, water and nutrient dynamics may well be possible. I hope funding can be made available to continue this important work.

Methods used were well described for both the EC measures, the experiment and a statistical analysis linking Singular Spectrum Analysis, Pearsons coefficient and information theory to identify key drivers of fluxes, plus lags between biophysical drivers of NEE all as a function of time and across the + N, +P and additive N and P treatments effects, impressive.

Given the quality of site and data set, the performance of the CT site is interest as it was a significant source to the atmosphere of ~80-100 g m2- y-1 or almost 1 t C per year lost from the system. Mean and interannual variability is similar to other semi-arid savanna as reported by Archibald et al (2009) for the semi-arid savanna at Kruger NP and Ma (2007) over oak savanna in California - more needs to made of this in the Discussion. Where is this carbon coming from – grazing and net loss from the soil? Can't be fire here. The word 'fire' is never mentioned in the paper, an oversight given this is a major feature of savanna ecosystems.

Revisions are required, overly long ms, somewhat verbose and repetitive, the writing needs to be tightened considerably! With revisions post-reviews will make for an excellent paper.

Technical comments

Introduction

Background provided in the Introduction was good

L52-55 "Light absorption ... Oritz (2024)" – consider delete these lines, your audience will be aware of this theory, basic plant physiology. There is a lot of these sort of statements in the Discussion as well.

L57 re-word – "Typical ecosystems in semi-arid regions are savannas where coexisting vegetation layers (e.g., tree and grass) interact in complex ways (Higgins *et al.,* 2000, House *et al.* 2003).

Cite some classic savanna ecology papers here to support this important claim, e.g.;

*Higgins, S. I., Bond, W. J., & Trollope, W. S. W. (2000). Fire, resprouting and variability: A recipe for grass-tree coexistence in savanna. Journal of Ecology, 88(2), 213–229. doi.org/10.1046/j.1365-2745.2000.00435.x*

*House, J. I., Archer, S., Breshears, D. D., Scholes, R. J., & Participants, N. T. I. (2003). Conundrums in mixed woody–herbaceous plant systems. Journal of Biogeography, 30(11), 1763–1777. https://doi.org/10.1046/j.1365-2699.2003.00873.x*

L60 delete "Especially the ..."

L61 should read "On the Iberian Peninsula..."

L66 "... ), limited by water in the dry season and by nutrients and energy in the wet season (Moreno, *et al.* 2008... }. Add Whitley et al 2011 here, relevant paper on light limitation on GPP in savannas

*Whitley, R., Macinnis-Ng, C., Hutley, L. B., Beringer, J., Zeppel, M., Williams, M., Taylor, D., & Eamus, D. (2011). Modelling productivity and water use across five years in a mixed C3 and C4 savanna using a soil-plant-atmosphere model: GPP is light limited not water limited. Global Change Biology, 17, 3130–3149. https://doi.org/doi.org/10.1111/j.1365-2486.2011.02425.x*

L84 delete "set up", replace with "established"

L110 "... 20-25 trees" a bit loose, do you have estimates of mean tree basal area in m2 ha-1 or similar tree size metric? Mean height also useful.

L112 spatially variability of grass - here and comment on significant seasonal temporal variability of grass growth that you describe L309 add text "described in detail below".

Giving an LAI range a single value not useful in this context

move Figure 1 here which is in the methods as more site information here would be useful.

You could also Fig 1 more comprehensive by adding 2 panels - Fig 1a) add a site map, this is lacking, showing location within country and the treatment locations, plus a second panel b) mean monthly precip and mean monthly Tair, and c) the current Fig of GCC of the grass layer over the growing season.

This will highlight the seasonality of this savanna climate system and the dynamic phenology, largely driven by the grass phenology, with presumably tree cover relatively constant.

L167 define the standard NDVI acronym "... and satellite data (normalized difference vegetation index, NDVI)".

L192 consider using Sentinal-2 EVI as well, or both indices. I have found with flux data that EVI covaries more closely than NDVI.

L311 "spring, dry down, summer, autumn and winter, as described above in Chapter 2.1.). Delete "as described above in Chapter 2.1), clearly a left over from your PhD thesis?!

L348 delete ".and".

L351 should read "For all plots, ..."

L401 "NT (12 days) compared to 15 and 16 days at NPT and CT,..." Would this be a significant difference? How would you test this?

In fact all the these rather modest differences in lags i.e. 2-4 days in this paragraph, are they significant or simply error / variability in the data?

L406-408 delete this text, focus on VPD as a driver.

L431 re-word "... phenological seasons based on the grass layer GCC derived from PhenoCam "

L435 re-word " as well as radiation parameters PAR and SWDR ..."

L443 re-word "Additionally SWDR, PAR were important in …".  There are numerous examples like this, a bit repetitive, inefficient writing.

L454 No need for this sub-heading, delete "3.5 Changes in NEE Sensitivity over Time"  seemed to me like this text is continuing description of Table 2.

L476 "Fig. 6", add a space

L485 delete "amount"

L515 see also Moore et al 2016 Biogeosciences 13: 5085-5102, doi:10.5194/bg-13-5085-2016.

L544 re-word "do not appear to play a crucial role at the seasonal scale."

L560 "growing season, spring, NEE is dominated by GPP." Careful making statements like this as these two variables are not independent of each other ie GPP is derived from NEE observations. You would have to be very confident of your Reco model used in this system to estimate GPP.

L592 delete "made"

---

## Author Comment (AC1)

| DOI | https://doi.org/10.5194/egusphere-2024-3190 |
| --- | --- |
| Altered Seasonal Sensitivity of Net Ecosystem Exchange to Controls Driven by Nutrient Balances in a Semi-arid Savanna | |
| Author(s) | Laura D. Nadolski et al. |
| Handling editor | Marijn Bauters |
| Manuscript type | Research article |
| Status | Authors response to reviewer |

**Response to Report from Reviewer #1**

Reviewer comments are printed in black.
Answers are printed in blue below the respective comment.
* * *
In a world dealing with a changing climate, there is a need for studies investigating environmental changes following anthropogenic influences, especially in understudied ecosystems with complex dynamics such as semi-arid savannas. This study uses an unique long term dataset collected in a large-scale nutrient addition experiment in a semi-arid savanna in Spain to look into the effect of altered nutrient levels on the relationships between NEE and it's key drivers, using robust methods as Singular Spectrum Analysis and Information Theory. The long term dataset is analyzed both as a whole and divided into phenological seasons, which results in a deeper understanding of the ecosystem as well as interesting insights into the effects of the nutrient addition, underneath the water or energy limitation during different seasons. The methods are well explained and the important results are well discussed, however some points require further clarification or discussion.

Thank you very much for this assessment of our work, for pointing out its relevance and for the very helpful comments. We considered all of them in detail and they have helped us to improve the overall quality and comprehensiveness of this manuscript. Please find below a point-by-point reply to your comments and suggested changes in the revised manuscript.

**Specific comments:**
*Materials and Methods*
Line 109: Authors could add a map of the region
Thank you for your suggestion, we agree it is great to have this. The Reviewer #2 also suggested adding a site map.
We have added a map with the location of the sites on the Iberian Peninsula and an airborne image showing the location of the three eddy covariance towers. Following suggestions from the Reviewer #2, we additionally compiled a plot showing the monthly mean precipitation sums and temperature across the study period (2016-2023).

[Figure]

"Fig.1: a) site location on the Iberian Peninsula. b) location of the three eddy covariance towers. Nitrogen added tower (NT) is in blue, control tower (CT) is in purple, and nitrogen + phosphorous added tower (NPT) is in light blue. The tower locations were chosen in a way that during dominant wind directions their footprints do not overlap. Footprint climatologies can be found in Fig.1 in El-Madany et al. (2018). c) average monthly precipitation sums and temperature (measured at 15m) across 2016-2023."

Line 137: Is there more information on when N and P were applied to the plots in terms of seasons or years? Would you suspect that the results over the years, for example in Fig 6, could be in anyway linked to the timing of the application of N and P?

We agree with the reviewer and have provided more detailed information on the fertilization scheme as follows:

"The N and P fertilization was applied around similar time at the sites each year, with some exceptions due to weather or logistics restrictions (e.g., pandemic). N was added at 100, 20, 50, 24 and 12 kg N ha$^{-1}$ at both sites by end of winter of 2015, 2016, 2017, 2021 and 2023, respectively, and P was added at 50, 10, 25, 6, 6 and 6 kg P ha$^{-1}$ at NPT in fall of 2014, 2015, 2016, 2019, 2020 and 2022, respectively. This timing of the application of N and P was selected to have maximal possibility to be used by vegetation in the next growing season after each addition."

As the timing of the application of N and P was chosen to increase the possibility to be used by vegetation in the next growing season, the increasing trend and altered variability of NEE at NT and NPT might be smaller if fertilization was applied at different timing. We have now added this potential uncertainty in the end of Section 4.4 as follows:

"As the timing of the application of N and P was chosen to increase the possibility to be used by vegetation in the next growing season, the observed changes in NEE and driver importance at NT and NPT might be smaller if fertilization was applied at different timing."

Line 154. How many soil sensors were installed per footprint? And were they installed in open field or under trees or both? There is mention in the discussion part that soil temperatures below oak trees are more important than those in open areas during the regreening in autumn (line 575) and that this also could be related to the variations in soil moisture between open and shaded pastures. Therefore it seems important to know where the sensors were located and if the authors were able to capture some of these variations in soil moisture that could underpin this statement.

Thank you for pointing out that clarifications are needed here.
We used sensors for soil temperatures and soil heat flux from two locations per tower: below the tree canopy and open pasture area. The ones below canopy are named with _Shd in the end (abbreviation for

shadow) and the ones in open pasture are suffixed with _Sun (specified in former lines 153-155). As the soil heat flux is influenced by the moisture content of the soil it allows assumptions on the variations in soil moisture between open pasture and shaded areas.

As the terms "shadow" and "sun" might not be the best choice, as below canopy it can be sunny and under open area it can be shadowed, we decided to change the suffx to _bc ("**b**elow **c**anopy") instead of "_Shd" and "_op" ("**o**pen **p**asture") instead of "_Sun".

While we originally had four soil moisture profiles per tower as well, there were many problems with the sensors over the past years, such as communication errors or malfunctions. Therefore, we used integrated soil water content values of the top 20 cm (normalized, in percent), which is weighted on the canopy cover of 20% to obtain soil water content values representative for the ecosystem. We have added this information to the manuscript as follows (former lines 153-156):

"Soil measurements comprised soil temperature in open pasture (Tsoil_op) and below oak tree canopy (Tsoil_bc) as well as soil heat flux in open pasture (SHF_op) and below oak tree canopy (SHF_bc). For soil water content, we used the different measurements integrated over the top 20 cm of the soil, weighted by a canopy cover of 20% to obtain soil water content values (SWCn) representative for the ecosystem."

The authors use only daytime measurements to calculate the aggregated daily means of NEE. Why is the nighttime data removed?

Thanks for your comment, we agree that clarification is necessary here. We used daily aggregated means of daytime NEE to only rely on reliable data.
We only use non-gapfilled, measured flux data, so the driver identification would not be confounded by gap-filling techniques based on meteorological measurements. We selected measured flux data with the highest quality (i.e. quality flag = 0, flagging policy following Mauder and Foken 2004). Therefore the data coverage of the half-hourly timeseries is quite low (around 30% on average), especially during the nighttime, as well-developed turbulence and stationary atmospheric conditions are oftentimes violated during nighttime hours (i.e., quality flag is not 0). Thus, including nighttime-values would introduce a substantial bias in the analysis, as some daily aggregates would have included a significant amount of nighttime data and most others are without nighttime data. We therefore aggregated daytime daily values. We have added the following information into the manuscript (section 2.4.1):

"For our analysis we calculated from the biometeorological and flux data daily mean values aggregated from the half-hourly measured values during daytime. We only use non-gapfilled, measured flux data, so that the driver identification is not confounded by gap-filling techniques based on meteorological measurements. To ensure that there are only high-quality measured values, we selected data with quality flag = 0 (flagging policy according to Mauder and Foken (2004)). Consequently, the data coverage of the measured half-hourly timeseries is quite low (around 30%) and especially heterogeneous during the nighttime. Therefore, we calculated from the biometeorological and flux data daily mean values by aggregating only daytime measurements to avoid the bias."

Line 228 – 230 "In the second step X is decomposed into its orthogonal components by determining eigenvalues and eigenvectors corresponding to principal components (singular value decomposition). Then the eigenvalues of the covariance matrix X · X are ranked."
I don't know the SSA method very well, however this part is slightly confusing for me as I think it is the eigenvalues and eigenvectors of XXᴛ that are determined and then X is decomposed in matrices with rank 1 which are constructed using these eigenvalues and eigenvectors. For me the term "decomposed in its

orthogonal components" sounds vague. Also maybe avoid the use of "ranked" as I think you mean ordered by decreasing magnitude here. Rank in terms of matrices can be confused with the terminology of rank of a matrix.

Thanks for your suggestions to clarify the description of the SSA method. We decided to not get into more detail regarding the method here, as thorough descriptions can be found in the literature (Golyandina and Korobeynikov, 2014; Golyandina and Zhigljavsky, 2013). The expression "decomposed into orthogonal components" is used in the literature (Baldocchi et al., 2021; Mahecha et al., 2010) and therefore we kept it.
However, we have rephrased the respective sentences for better clarity (former lines 228-230):

"In the second step a singular value decomposition of X is performed and it is decomposed into its orthogonal components by determining eigenvalues and eigenvectors corresponding to principal components. The eigenvalues of the covariance matrix X · X are then ordered in decreasing magnitude."

Line 239: Here is stated that the data is gap-filled using the igapfill – function, however in Line 205 – 206 you state that you only use measured values to avoid confounding with the other meteorological variables in later analyses. Is this gap filling necessary because the method requires a full time series?
Yes, you are completely right, the gap-filling is necessary because the singular spectrum analysis requires a time series without gaps. We indicated that as follows in the respective section:
"First, as required by SSA, we gap-filled the timeseries with the rssa package's internal function, igapfill, which fill gaps using the low-frequency component of the timeseries itself (i.e., not based on meteorological measurements)."

Will you not introduce this confounding again by gap filling here, in the sense that even though you remove the gap filled values again after the SSA (mentioned in Line 299), the gap filling will have an influence on the SSA result? Or is this method of gap filling not based on the meteorological data?
Thanks for expressing your concern on the gap-filling.
The gap-filling with the igapfill()-function does not confound the results, as it is not based on meteorological data or any other ancillary data, but solely on the low-frequency component of the timeseries itself, as stated at Lines 222-224. But, to make it clear, we have edited the sentence as shown in the previous response.

Section 2.4.4: I think that there are some inconsistencies with equations here. Formula (4) has no double sum, iterating over both x and y. This double sum is however seen in equation (6). Equation (6) on the other hand has no logarithm included in the right hand side of the equation both in the numerator as in de denominator (as is present in equation (4)). Also a maximum operator should be present as $MI_{max}$ is the maximum iterated over different values of tau and in the denominator of equation (6) an "i" pops up. Maybe check that these formulas are indeed consistent and correct.

Thank you for pointing out inconsistencies in the equations. Indeed, the double sum in equation (4) was missing, we have corrected it as follows:

$$MI = \sum_{x_t, y_t} p(x_t, y_t) log_2 \frac{p(x_t, y_t)}{p(x_t)p(y_t)}$$

And the logarithms were missing in the notation of Equation (6), we have added them now, and switched the accidental "i" for a "t". In addition, a maximum operator was introduced. The revised Equation (6) is now as follows:

$$MI_{max} = MI_{sync_{(\tau)}} = \max\left(\frac{\sum_{x_{t-\tau}}\sum_{y_t} p(x_{t-\tau}, y_t)\log 2\frac{p(x_{t-\tau}, y_t)}{p(x_{t-\tau}), p(y_{t})}}{-\sum_{y_t} p(y_t)\log 2\; p(y_t)}\right)$$

*Results*
Section 3.2.

Line 351 – 354 states the common, most important predictors which are nicely highlighted and explained in the discussion, section 4.2
Line 385 – 388 highlights that r and MI nicely agreed in the detection of the most important drivers, which explains why you continue only with the MI measurements.

Line 355 – 384 describes the key controls for each plot, however I would suggest to make this shorter and more to the point. It reads difficult due to the many variable names and these separate results for each plot are nowhere discussed in the discussion section. The comparison between the towers in section 3.3 contributes more to the story of the paper than these separate observations, in my opinion. The section and graph are needed to support discussion section 4.2 but I feel like the separate observations could be either shortened or restructured.

We appreciate this comment. We have addressed this comment in the results section 3.2 by revising the respective paragraphs and shorten them to make it more concise.

"At CT, T$_{soil}$_bc and Ta15 further exhibited strong interactions with NEE using both *r* and MI$_{sync}$ (Fig.3 (a), (b)). Variables describing water availability, such as VPD, SWCn and Rh were ranked in the middle ranges by MI$_{sync}$. The MI analysis provided deeper insights into the interactions between the environment and NEE by considering leading and lagging effects, as shown by MI$_{max}$ (Fig. 3(c)). NDVI showed the highest interaction with NEE at a time lag of 16 days, and gcc_gr had a lag of 7 days. When considering leading and lagging effects, EF became relatively less important. Soil temperatures were identified amongst the five most important controls. SWCn was also important with a 20-day lag. Other variables such as air temperature and VPD showed the highest interaction with a lag of around a month. Radiation-related variables like PAR and SWDR exhibited long lag times in their highest interaction with NEE (60 days and 53 days, respectively). All MI values can be found in the Supplementary Material (S3).
At NT, soil temperatures, VPD, SWCn and air temperatures were among the most significant controls identified by both synchronous methods, following vegetation greenness and EF. NDVI showed the highest interaction with NEE with a lag of 12 days, followed by gcc_gr with a lag of 6 days. Soil temperatures exhibited the highest interactions with a lag of around a month, while air temperatures showed the highest interaction at a lag of 26 days. Moisture-related variables all showed similar time lags (16-20 days). EF had the highest interaction with NEE at a lag of two weeks. Shortwave radiation-related variables showed a strongly lagged effect (i.e., PAR 59 days, SWDR 57 days) (Fig.3 (f)).
At NPT, both *r* and MI$_{sync}$ detected soil temperatures, air temperatures and VPD as the most important NEE controls behind gcc_gr and NDVI (Fig.3 (g), (h)). NDVI and gcc_gr led NEE with the strongest interaction

at lags of 2 weeks and 10 days, respectively, followed by soil temperatures and air temperatures with the highest interaction at a lag of around a month (Fig.3 (i)). EF showed the highest interaction at a lag of 12 days. Other moisture-related variables like VPD, SWCn, and Rh were also detected to be in the middle ranks by $MI_{max}$, with time lags of 20-26 days. PAR and SWDR showed the highest interaction with NEE at time lags of around 50 days (Fig.3 (i)). "

Table 2: the authors might add the MI values in the table or in supplementary material of the five most important drivers for each phenological season. Would be interesting to see if for example the three first ones have a way higher MI value than the two last ones, or if the values are all close to each other.

Thank you, we agree that it is interesting for the reader to see if the MI values are close to each other or not. Therefore, we have added the respective $MI_{sync}$ values with two decimal places in brackets in Table 2.

*Discussion.*
Section 4.1

Line 475: there is an increasing trend in the difference between annual NEE maximum and annual NEE minimum in the NPT plot. Is both the minimum value going down and the maximum value going up? Or is this increase in difference guided by mainly one of the two?
Thanks for your comment. The increases in maximum values are a bit stronger than the decreases in minimum values. The rising difference is therefore slightly more driven by the maximum value. However, the interannual variability of the maximum and minimum values is high. We think this information is a nice addition and have added it to section 4.1 in the revised manuscript as follows:

"This trend is driven slightly more by increasing maximum values."

Only NPT has a significant increase, however the explanation as why this could be the case, does not specifically mention only N+ P addition, so you would also expect this increase in the N plot. Is there an explanation why this is not the case? Or would you nuance the p-value and suggest that this increase might also be the case for N addition?
Yes, increase is also the case for the N addition site. We have edited the sentence to be clearer as follows (former lines 473-477):
"Our results indicate that both nutrient addition cases enhance seasonal NEE variability compared to the control. Additionally, the seasonal variability increases over time at both fertilized plots. Looking at the difference between annual NEE maximum and annual NEE minimum, we notice substantial increasing trends at both site, with the trend at NPT plot being significant (Fig.6)."

Line 530: here is stated that EF, which is a proxy for rain pulses, is an important driver but that SWC, VPD and RH are not important drivers following the MI analyses. This seems to be contradictory results as you mention both of them as proxies for rainfall. Points this towards the fact that in this study soil water related variables and rain pulses are not as important as previously found in other studies and that the link with EF as important driver is more based on its relation with LAI or radiation than with moisture related variables? However in line 544 is also stated that Radiation does not seems to have a major influence. How can this opposite results be interpreted?

Thank you for pointing this out. We agree that the results here seem a bit contradictory and we were also debating about this.

We argue that indeed the rain pulse effect plays an important role in this ecosystem; however, its importance is limited to the dry summer months, whereas in the regreening season, winter and spring, water availability is abundant. Therefore, in the respective plots (Fig. 3 & Fig. 4), which depict the data of all years across the whole year (including the seasons with high water abundance), the importance of the rain pulse effect in explaining NEE diminishes. Therefore, in this context the relation of EF with LAI might be the dominant one. We clarified it in the manuscript (former line 532) to avoid confusion:

"This might point to the relationship of EF with LAI being the dominant one in this context, as the vegetation indices are higher in their importance than other water related variables."

Section 4.3: the authors could restructure the sections to follow the order of Table 2 and section 3.4. or restructure the table and the accompanying result section to follow the order from the discussion section.

Thank you for your suggestion, we agree that changing the order of the section according to the structure of Table 2 and section 3.4 facilitates reading and understanding this part of the discussion.
We have made the respective adjustments in section 4.3 as follows:

[revised manuscript text omitted]

**Technical corrections:**
*Abstract*
Line 15: Semi-arid ecosystems dominate the variability and trend of the terrestrial carbon sink.
Thank you, we have corrected that.

Line 29 -30: The increasing NEE variability might become even more pronounced with increasing N deposition and a changing climate in the future.
Thanks, we have corrected it.

*Introduction*
Line 62: human shaped man-made savanna-like agroecosystem, …
Thanks for your suggestion, we have corrected it.
Line 72-73: Few studies so far have dealt so far with …
Thank you, we have corrected that.

*Materials and Methods*
Table 1: add in the caption that the soil heat flux and soil temperature in the shadow were calculated based on the shadow fraction … Also the height of the air pressure device is not present
Thank you for your comment. The soil heat flux and soil temperatures were not calculated based on shadow fractions, but were measured indeed in the shadow (_Shd, under tree canopy) and in the sun (_Sun, in open field). However, we have added this calculation information for SWCn as follows:
"normalized soil moisture content for top 20cm using the shadow fraction of 20% to represent ecosystem values"

Also, we have added the heights of the air pressure devices to table 1 (15m, 15.5m (CT).

Line 113: heterogeneous with values between 0.5 and 2.5 ….
Thank you, we have corrected that.
Line 151: here CO2- flux is also in the list of additional atmospheric variables, maybe remove
Thank you, we have removed it.
Line 173: and collected red, blue, ….
Thanks, we have corrected that.
Line 291: Positive and negative values of $\tau$ show an asynchronous interaction between X and Y, with a lag …
This phrasing may sounds like it insinuates that it is the value of tau that is the interaction, however tau is merely the time lag. Maybe rephrase.
Thank you for pointing this out, we have rephrased the sentence for more clarity as follows:
"When τ is positive or negative (= 0), the interaction between X and Y is characterized as asynchronous, with τ showing the lead or lag in Y relative to X, respectively."

*Results*
Line 347-350: this sentence is difficult to understand and has a point which seems wrongly placed. "accounts for collinear relationships. and MIsync and …"
Thank you for your comment. We have rephrased the sentences and changed their structure, to provide more clarity and correct grammar as follows:

"Pearson correlation coefficient r considers only linear relationships between variables; Mutual Information (MI), accounts for collinear relationships. MI$_{sync}$ and r values show synchronous relationships, MImax values can account for leading and lagging interactions by identifying the day of the highest interaction between the potential driver and NEE within a 60-day window."

Line 411: remove the sentence about soil temperature here, soil temperature is again mentioned in a better way in Line 416
Thanks, we have removed that sentence.

Line 455: "We observed that with N addition, NEE became less sensitive to certain variables during autumn (i.e., the regreening phase), the drydown phase, and winter over time (Fig.5)."
In winter this is "more sensitive" and not "less sensitive" as there is increase as mentioned in line 459.
Thank you for pointing this out, we have corrected that and removed "and winter".

Figure 5: the caption states that NT vs CT is in the bottom figure and NPT vs CT in the top figure but it is the way around.
Thank you very much for spotting this, we have corrected it.

---

## Author Comment (AC2)

| DOI | https://doi.org/10.5194/egusphere-2024-3190 |
|---|---|
| Altered Seasonal Sensitivity of Net Ecosystem Exchange to Controls Driven by Nutrient Balances in a Semi-arid Savanna | |
| Author(s) | Laura D. Nadolski et al. |
| Handling editor | Marijn Bauters |
| Manuscript type | Research article |
| Status | Authors response to reviewer |

**Response to Report from Reviewer #2**

Reviewer comments are printed in black.
Answers are printed in blue below the respective comment.
* * *
This is a sound piece of work showcasing the use of eddy covariance measures of ecosystem fluxes as an experimental tool, with measures examining impacts of N and P additions to semi-arid savanna ecosystem. Tree-grass savannas occupy ~30% of the global land surface and quantifying fluxes from these systems is critical as they appear to be a source at present to the atmosphere and they drive global variability of global atmospheric CO2. This is a unique experiment, and the EC work described here is a great add on, highlighting the power of long-term studies - another 7 or 8 years we will start to pick up impacts of CO2 fertilisation, heatwaves, increasing variability of precip, as well as the +N and P effects reported. Given the novel statistical analytics described in this study, untangling all of these interacting drivers of C, water and nutrient dynamics may well be possible. I hope funding can be made available to continue this important work.
Methods used were well described for both the EC measures, the experiment and a statistical analysis linking Singular Spectrum Analysis, Pearsons coefficient and information theory to identify key drivers of fluxes, plus lags between biophysical drivers of NEE all as a function of time and across the + N, +P and additive N and P treatments effects, impressive.
Given the quality of site and data set, the performance of the CT site is interest as it was a significant source to the atmosphere of ~80-100 g m2- y-1 or almost 1 t C per year lost from the system. Mean and interannual variability is similar to other semi-arid savanna as reported by Archibald et al (2009) for the semi-arid savanna at Kruger NP and Ma (2007) over oak savanna in California - more needs to made of this in the Discussion. Where is this carbon coming from – grazing and net loss from the soil? Can't be fire here. The word 'fire' is never mentioned in the paper, an oversight given this is a major feature of savanna ecosystems.
Revisions are required, overly long ms, somewhat verbose and repetitive, the writing needs to be tightened considerably! With revisions post-reviews will make for an excellent paper.

Thank you very much for your kind assessment of our work, for suggesting improvements and proposing important additional literature. We have shortened the Methods and the Results section considerably to make the overall manuscript more concise, complemented the Discussion section as suggested and addressed all technical comments in detail. Your suggestions have helped us to improve the quality of this manuscript in various aspects.
Please find below a point-by-point reply to your comments and suggested changes in the revised manuscript.

We do not mention fire in the discussion as wildfires happen rarely in such managed ecosystems in Spain, and in the concrete case of our study site, no fire happened as far as we know and as far as local people remember, which mean at least during the last 50 years.

In the case of our site, the long-term NEE estimates from EC data (measurements since 2004) suggest the ecosystem acts a significant source of carbon, but they are challenged by the estimates of carbon stocks performed at the site (tree biomass stock changes from dendrometers and soil inventories performed in 2006, 2015 and 2019), which suggest that the carbon budget of the ecosystem is rather neutral. The analysis and discussion of long-term carbon budget at the site is another work in progress to be concluded and published in the near future. Therefore, despite the measured annual NEE for the analyzed period is about 90 $gCm^{-2} y^{-1}$, there is so far not undisputable evidence that the ecosystem acts as a significant source of carbon. As no clear statement about the mean carbon balance of the ecosystem can be made now, and as it is not in the scope of the presented study, we prefer not to speculate about possible carbon sources.

We added the following paragraph in the beginning of section 4.1 to the discussion, aligning with your suggestions to compare with mean and interannual variability of other semi-arid savannas:

"The annual NEE average from EC measurement is about 90 $gC\ m^{-2} y^{-1}$, suggesting the unfertilized site acts a carbon source with a high interannual variability. Similar mean and variability were found in other semi-arid savannas, such as in Kruger National Park in South Africa (75 ±105 $gC\ m^{-2} y^{-1}$) (Archibald et al., 2009). However, semi-arid savannas can also act as carbon sinks. In California, a similar oak savanna (i.e., Tonzi Ranch) was observed to be a carbon sink (values from -144 to -35 $gC\ m^{-2} y^{-1}$), while the neighboring grassland (i.e., Vaira Ranch) was found to be a carbon source (-88 to 189 $gC\ m^{-2} y^{-1}$)(Ma et al., 2007). In Dakar, Senegal, a Sahelian savanna ecosystem acted as a carbon sink with an average annual NEE budget of $-180 \pm 29\ g\ C\ m^{-2} y^{-1}$ (Wieckowski et al., 2024). Another natural West-African savanna in the South of Burkina Faso has been found to be a strong sink of $CO_2$ ($-864$ to $-1299\ g\ CO^2\ m^{-2} y^{-1}$) while two degraded sites nearby were $CO_2$ sources (118 to 605 $g\ CO^2\ m^{-2} y^{-1}$)(Berger et al., 2019)."

**Technical comments**
Introduction

Background provided in the Introduction was good
Thank you.

L52-55 "Light absorption … Oritz (2024)" – consider delete these lines, your audience will be aware of this theory, basic plant physiology. There is a lot of these sort of statements in the Discussion as well.
Thanks for pointing this out, we have deleted the respective lines and also took your comment into account while revising the Discussion section.

L57 re-word – "Typical ecosystems in semi-arid regions are savannas where coexisting vegetation layers (e.g., tree and grass) interact in complex ways (Higgins *et al.,* 2000, House *et al.* 2003).
Cite some classic savanna ecology papers here to support this important claim, e.g.;
*Higgins, S. I., Bond, W. J., & Trollope, W. S. W. (2000). Fire, resprouting and variability: A recipe for grass-tree coexistence in savanna. Journal of Ecology, 88(2), 213–229. doi.org/10.1046/j.1365-2745.2000.00435.x*
*House, J. I., Archer, S., Breshears, D. D., Scholes, R. J., & Participants, N. T. I. (2003). Conundrums in mixed woody–herbaceous plant systems. Journal of Biogeography, 30(11), 1763–1777.*
*https://doi.org/10.1046/j.1365-2699.2003.00873.x*

Thank you for your comment and for suggesting further references. We have added them to our bibliography and rephrased the sentence in the following way:

"In semi-arid regions, savannas are a typical ecosystem type. They comprise coexisting vegetation layers (e.g., tree and grass), which interact in complex ways (Higgins et al., 2000; House et al., 2003)."

L60 delete "Especially the …"
Thanks, we have deleted that.

L61 should read "On the Iberian Peninsula…"
Thank you, we have corrected that.

L66 "… ), limited by water in the dry season and by nutrients and energy in the wet season (Moreno, *et al.* 2008… }. Add Whitley et al 2011 here, relevant paper on light limitation on GPP in savannas
*Whitley, R., Macinnis-Ng, C., Hutley, L. B., Beringer, J., Zeppel, M., Williams, M., Taylor, D., & Eamus, D. (2011). Modelling productivity and water use across five years in a mixed C3 and C4 savanna using a soil-plant-atmosphere model: GPP is light limited not water limited. Global Change Biology, 17, 3130–3149. https://doi.org/doi.org/10.1111/j.1365-2486.2011.02425.x*
Thank you for recommending this important reference, we have added it.

L84 delete "set up", replace with "established"
Thanks, we have replaced that.

L110 "… 20-25 trees" a bit loose, do you have estimates of mean tree basal area in m2 ha-1 or similar tree size metric? Mean height also useful.
Thank you for pointing out that this information is missing. We have added it to the respective sentence as follows:
"The tree density is around 20-25 trees per hectare, with a mean diameter at breast height of 46 cm (El-Madany et al., 2018), the fractional canopy cover of trees is 23 % and the canopy height is on average 8.7 m (Bogdanovich et al., 2021)."

L112 spatially variability of grass - here and comment on significant seasonal temporal variability of grass growth that you describe L309 add text "described in detail below".
Giving an LAI range a single value not useful in this context
Thank you for your comment, we have revised the text as follows:

"The tree leaf area index (LAI) is around 0.35 $m^2$ $m^{-2}$, the grass layer has a peak LAI in spring but is quite spatially variable with values between 0.50 and 2.50 $m^2$ $m^{-2}$ due to the seasonal temporal variability of grass growth (described in detail below) (Migliavacca et al., 2017)."

move Figure 1 here which is in the methods as more site information here would be useful.
You could also Fig 1 more comprehensive by adding 2 panels - Fig 1a) add a site map, this is lacking, showing location within country and the treatment locations, plus a second panel b) mean monthly precip and mean monthly Tair, and c) the current Fig of GCC of the grass layer over the growing season. This will highlight the seasonality of this savanna climate system and the dynamic phenology, largely driven by the grass phenology, with presumably tree cover relatively constant.

Thank you for the suggestions on the figures. We have added a site map (which was also suggested by reviewer #1), indicating the location of the sites on the Iberian Peninsula as well as the location of the three eddy covariance towers on an airborne image. Following your suggestion, we have further added a figure depicting the monthly mean temperature and precipitation sums over the study period (2016-2023):

[Figure]

"Fig.1: a) site location on the Iberian Peninsula. b) location of the three eddy covariance towers. Nitrogen added tower (NT) in blue, control tower (CT) in purple, nitrogen + phosphorous added tower (NPT) in light blue. The tower locations were chosen in a way that during dominant wind directions their footprints don't overlap. Footprint climatologies can be found in El-Madany et al. (2018), Fig.1. c) average monthly precipitation sums and temperature (measured at 15m) across 2016-2023."

We added it in the suggested position, at the end of the section "Site description". As this figure already comprises three panels, we decided to keep the figure with GCC of the grass layer separate and in the "Phenological Seasons"-section.

L167 define the standard NDVI acronym "… and satellite data (normalized difference vegetation index, NDVI)".
Thank you, we have defined the acronym here.

L192 consider using Sentinal-2 EVI as well, or both indices. I have found with flux data that EVI covaries more closely than NDVI.
Thank you for the suggestion, you make a valuable point and we discussed including EVI from the FluxnetEO dataset (MODIS) as an additional variable. However, NDVI is already the most important driver, showing the highest mutual information and Pearson correlation values. Therefore, after some consideration we decided to not include EVI, as it would probably show a similar pattern as NDVI with higher interaction values and it will likely not provide additional insights. Nonetheless we agree that EVI is an important vegetation index and we mention its potential in the discussion (4.3) (former line 629):

"Alternatively to NDVI, the Enhanced Vegetation Index (EVI) could be considered as a representation of vegetation greenness, as it is found to covary closely with the carbon flux in semi-arid ecosystems (Maluleke et al., 2024)."

L311 "spring, dry down, summer, autumn and winter, as described above in Chapter 2.1.). Delete "as described above in Chapter 2.1), clearly a left over from your PhD thesis?!
Thank you for pointing this out, we have deleted it.

L348 delete ".and".

Thank you, we have corrected that and revised the sentence according to the comments of reviewer #1 in the following way so that everything is grammatically correct:

"Pearson correlation coefficient $r$ considers only linear relationships between variables; Mutual Information (MI), accounts for collinear relationships. $MI_{sync}$ and $r$ values show synchronous relationships, $MI_{max}$ values can account for leading and lagging interactions by identifying the day of the highest interaction between the potential driver and NEE within a 60-day window."

L351 should read "For all plots, …"
Thank you, we have corrected that.

L401 "NT (12 days) compared to 15 and 16 days at NPT and CT,…" Would this be a significant difference? How would you test this?
In fact all the these rather modest differences in lags i.e. 2-4 days in this paragraph, are they significant or simply error / variability in the data?
Thank you very much for pointing this out, this is a valuable point. While we tested and know that the results of each tower are significant, the significance of the tower differences in MI values and lag times is not proved. As our paper aims to show general patterns of the effect of nutrient addition on NEE driver importance, we decided to not put the focus on the differences in the specific values and revised section 3.3 in a way that it highlights the overall patterns:

"N fertilization appeared to shorten the reaction time of NEE to changes in NDVI. GCC at the grass level showed higher explanatory value for NEE at NPT and NT ($MI_{max} = 0.37$) compared to CT ($MI_{max} = 0.33$). EF showed only slight differences in interaction strengths among the sites (Fig. 4, S3). Relative humidity at two heights showed the lowest interaction with NEE at CT ($MI_{max} = 0.24$), while the fertilized sites had slightly higher values (0.26-0.27). The reaction time of NEE to relative humidity appeared to decrease with fertilization. VPD appeared to have the higher explanatory value for NEE at NT and NPT, and slightly less and CT ($MI_{max} = 0.27$). The interaction between NEE and air temperatures was slightly higher at the fertilized plots compared to the control. Soil temperatures showed similar interaction strength with NEE across treatments, with $MI_{max}$ ranging from 0.31 to 0.33 (Fig. 4, S3).
Regarding radiation variables, PAR seemed to have a slightly higher interaction with NEE at NT, than at NPT and CT. Similarly, SWDR showed slighlty higher interaction with NEE at NT, while at NPT and CT it was equally strong.
In terms of soil variables, soil temperatures exhibited the strongest interaction with NEE. While soil temperatures below the canopy ($T_{soil}\_bc$) were almost the same across sites ($MI_{max} = 0.33$), the importance of soil temperatures under open air were lower at CT compared to the fertilized plots. SWCn showed the highest explanatory value for NEE at NT (Fig. 4, S3). An overview plot with all variables including the ones with $MI_{max} < 0.2$ is provided in the Supplementary Material (S4)."

L406-408 delete this text, focus on VPD as a driver.
Thank you, we considered deleting this part and have edited it within the revision of chapter 3.3 as shown in the previous answer.

L431 re-word "… phenological seasons based on the grass layer GCC derived from PhenoCam "
Thanks for your suggestion, we have revised the sentence in the following way:

"We split the 7-year dataset into five different phenological seasons based on grass layer GCC, and calculated $MI_{sync}$ between NEE and each of the drivers."

We have removed "derived from PhenoCam photos" as this was already explained above and redundant at this point.

L435 re-word "as well as radiation parameters PAR and SWDR …"
Thank you, we have rephrased that sentence as suggested.

L443 re-word "Additionally SWDR, PAR were important in …". There are numerous examples like this, a bit repetitive, inefficient writing.
Thank you, we have edited this part as suggested.

L454 No need for this sub-heading, delete "3.5 Changes in NEE Sensitivity over Time" seemed to me like this text is continuing description of Table 2.
Thanks a lot for your comment. We understand your point but after some consideration we would prefer to keep the sub-heading here, as the previous chapter targets a different analysis. While section 3.4 shows the results of the analysis of different NEE controls in different phenological seasons, section 3.5 deals with the development of their importance over time. To make this distinction clearer we think it is advantageous for a facilitated comprehension to keep the sub-heading here. But we have added a short sentence in the beginning to emphasize the difference as follows:
"Using yearly MIsync for each single season, we observed that with N addition, NEE became less sensitive to certain variables during autumn (i.e., the regreening phase), the drydown phase, and winter over time (Fig.5)."

L476 "Fig. 6", add a space
Thank you, we have added a space here.

L485 delete "amount"
Thanks, we have deleted that.

L515 see also Moore et al 2016 Biogeosciences 13: 5085-5102, doi:10.5194/bg-13-5085-2016.
Thank you, we have added that reference.

L544 re-word "do not appear to play a crucial role at the seasonal scale."
Thanks, we have corrected that.

L560 "growing season, spring, NEE is dominated by GPP." Careful making statements like this as these two variables are not independent of each other ie GPP is derived from NEE observations. You would have to be very confident of your Reco model used in this system to estimate GPP.
Thank you for pointing this out. We have rephrased the sentence in the following way:
"In the primary growing season, spring, NEE is typically dominated by GPP."

L592 delete "made"
Thanks, we have deleted that.

---

## Referee Report (RR1)

The authors clearly took the advice and comments of the reviewers into account. The restructuring of the text improved the readability a lot. The adaptations in sections 3.2 and 3.3 made the text more concise. The questions that came up during the reviewing process were answered nicely and short additional explanations were added into the article in a precise and to the point manner.

Some small additional remarks:

Line 164: friction velocity is here shortened as ustar (also in the table) while in line 161 it is shortened as u*, make this consistent.

Section 2.2: it is not entirely clear for me what the authors mean with the caption of table 1

> Caption in table 1: Soil heat flux and soil temperatures were calculated based on the shadow fraction estimated from the solar zenith angle (variable SZA) and a canopy cover of 20%.

Therefor it is also not entirely clear which soil measurements are actually measured and which are calculated based on shadow fractions and/or canopy covers. I think it is useful to refine Line 166-169, stating clearly how many sensors are used for Tsoil_op, Tsoil_bc, SHF_op, SHF_bc and for SWCn and on which locations (under canopy or open field).

> Line 166 – 169: Soil measurements comprised soil temperature in open pasture (Tsoil_op) and below oak tree canopy (Tsoil_bc) as well as soil heat flux in open pasture (SHF_op) and below canopy (SHF_bc). For soil water content we used the different measurements integrated over the top 20 cm of the soil, weighted by a canopy cover of 20 % to obtain soil water content values (SWCn) representative for the ecosystem.

Line 183: here RGB is mentioned but only in line 185 the letters are explained.

Section 2.4.1: revise this: several sentences are added but now some information is mentioned twice.

Section 2.4.2: some mathematical details

Line 247: I think it is not X . X but X . $X^T$ or X . X'

Formula (6) is not entirely correct or completely clear, maybe rewrite as:

$$MI_{max} = \max_{\tau}\left(MI_{sync}(\tau)\right) = \max_{\tau}(\dots)$$

---

## Author Response (AR2)

| DOI | https://doi.org/10.5194/egusphere-2024-3190 |
|---|---|
| Altered Seasonal Sensitivity of Net Ecosystem Exchange to Controls Driven by Nutrient Balances in a Semi-arid Savanna | |
| Author(s) | Laura D. Nadolski et al. |
| Handling editor | Marijn Bauters |
| Manuscript type | Research article |
| Status | Authors response to technical corrections |

**Response to Technical corrections**

The reviewer's report is printed in black.
Answers by the authors are printed in blue.
* * *
The authors clearly took the advice and comments of the reviewers into account. The restructuring of the text improved the readability a lot. The adaptations in sections 3.2 and 3.3 made the text more concise. The questions that came up during the reviewing process were answered nicely and short additional explanations were added into the article in a precise and to the point manner.

Thank you very much for this assessment.

Some small additional remarks:

Line 164: friction velocity is here shortened as ustar (also in the table) while in line 161 it is shortened as u*, make this consistent.

Thanks for this remark, we changed both occurrences of "u*" into "ustar" in line 161.

Section 2.2: it is not entirely clear for me what the authors mean with the caption of table 1 Caption in table 1: Soil heat flux and soil temperatures were calculated based on the shadow fraction estimated from the solar zenith angle (variable SZA) and a canopy cover of 20%.

Therefor it is also not entirely clear which soil measurements are actually measured and which are calculated based on shadow fractions and/or canopy covers. I think it is useful to refine Line 166-169, stating clearly how many sensors are used for Tsoil_op, Tsoil_bc, SHF_op, SHF_bc and for SWCn and on which locations (under canopy or open field).

Line 166 – 169: Soil measurements comprised soil temperature in open pasture (Tsoil_op) and below oak tree canopy (Tsoil_bc) as well as soil heat flux in open pasture (SHF_op) and below canopy (SHF_bc). For soil water content we used the different measurements integrated over the top 20 cm of the soil, weighted by a canopy cover of 20 % to obtain soil water content values (SWCn) representative for the ecosystem.

Thank you for pointing out this inconsistency. Indeed, we made a small mistake in the caption of table 1, as only soil water content is calculated based on the shadow fraction estimated from the solar zenith angle and a canopy cover of 20%. We changed the caption of Table 1 as follows (lines 208-210):

"Table 1: Flux, meteorological, soil variables and vegetation indices used in this study. Soil water content was calculated based on the shadow fraction estimated from the solar zenith angle and a canopy cover of 20%."

For both soil heat flux and soil temperature there are two sensors each in open pasture ("_op") and below canopy ("_bc") (, i.e. two soil temperature sensors under open pasture, and two soil heat flux sensors under open pasture, and the same for below canopy). To be more precise we altered the lines 166-171 as follows:

"Soil measurements comprised soil temperature measured from two sensors in open pasture (Tsoil_op) and below oak tree canopy (Tsoil_bc), and soil heat flux from two sensors in open pasture (SHF_op) and below canopy (SHF_bc). We used the average of two sensors when both were available and otherwise the measured values of one sensor. Additionally, to calculate soil water content we used the different measurements integrated over the top 20 cm of the soil, weighted by a canopy cover of 20 % to obtain ecosystem soil water content values (SWCn).

Line 183: here RGB is mentioned but only in line 185 the letters are explained.

Thank you for pointing this out. We removed the "RGB" in line 183, as it is not necessary to specify at this point and as the details are explained in line 185-186. The respective section now reads as follows:

"We used daily mean GCC values extracted from images collected every 30 minutes by digital cameras (Stardot NetCam 5MP) which were installed at the top of each ecosystem EC tower facing north. The cameras were set up according to the protocol of the PhenoCam network (https://phenocam.sr.unh.edu/webcam/tools/) and collected red, blue, green (RGB) images (Luo et al., 2018)."

Section 2.4.1: revise this: several sentences are added but now some information is mentioned twice.

Thank you for your comment. We revised section 2.4.1 and removed the sentences in line 216-217 and in line 223-224 to avoid repetition and the section now reads as follows:

"To make sure that the driver identification is not confounded by gap-filling techniques based on meteorological measurements, we only use non-gapfilled, measured flux data. To ensure that there are only high-quality measured values, we selected data with quality flag = 0 (flagging policy according to Mauder and Foken (2004)). Consequently, the data coverage of the measured half-hourly timeseries is quite low (around 30%) and especially heterogeneous during the night-time. Therefore, we calculated from the biometeorological and flux data daily mean values by aggregating only daytime measurements to avoid the bias. Daytime includes only values measured after sunrise and before sunset, identified using the suncalc package in R (Thieurmel, 2017). This does not apply to vegetation indices as they were calculated as described above. GPP and $R_{eco}$ were not assessed in this study as partitioning methods depend on other environmental factors that would also confound the analysis of NEE controls.
If not stated differently, the following analyses cover the 7-year period from 2016-2022 as in this time all variables are available. For the assessment of NEE variability and budgets, we utilized data spanning 8 years (2016-2023) because this extended dataset was available and incorporating additional years enhances the robustness of observed trends."

Section 2.4.2: some mathematical details
Line 247: I think it is not X . X but X . XT or X . X'
Thanks, we corrected that and changed it to X . X'.

Formula (6) is not entirely correct or completely clear, maybe rewrite as:
$MImax$ = max$\tau$ ($MIsync(\tau)$) = max$\tau$ (... )
Thank you for your remark. $MI_{max}$ identifies the maximum value and not the maximum time lag, therefor we believe that the notation is correct as it is. However, we noticed that $\log_2$ was not noted correctly and we corrected that as follows:

$$MI_{max} = max(MI_{sync_{(\tau)}}) = max(\frac{\sum_{x_{t-\tau}}\sum_{y_t} p(x_{t-\tau}, y_t) \log_2 \frac{p(x_{t-\tau}, y_t)}{p(x_{t-\tau}).p(y_t)}}{-\sum_{y_t} p(y_t) \log_2 p(y_t)})$$                         (6)

Thank you very much for contributing and helping us to improve this manuscript!

While working on the manuscript, we realized that the numeration of the figures is not correct anymore, since we added a new figure (Fig.1) with a map and climate diagram during the first revision process. We corrected the numeration of the figures in their captions and in the text.

Further, we noticed that in the supplementary plots the soil variables (soil temperature and soil heat flux) still had the old suffices ("_Shd" and "_Sun" instead of "_bc" and "_op"). We corrected that in the supplementary figures (S1.1, S1.2, S4 and S5) accordingly.

Laura Nadolski